

# MAP-IO, an atmospheric and marine observatory program onboard $MarionDufresne$ over the Southern Ocean

Pierre Tulet[1], Joel Van Baelen[2], Pierre Bosser[3], Jérome Brioude[2], Aurélie Colomb[4], Philippe Goloub[5], Andrea Pazmino[6], Thierry Portafaix[2], Michel Ramonet[7], Karine Sellegri[4], Melilotus Thyssen[8], Léa Gest[2], Nicolas Marquestaut[9], Dominique Mékiès[2], Jean-Marc Metzger[9], Gilles Athier[1], Luc Blarel[5], Marc Delmotte[7], Guillaume Desprairies[9], Mérédith Dournaux[1], Gaël Dubois[5], Valentin Duflot[2], Kevin Lamy[2], Lionel Gardes[10], Jean-François Guillemot[9], Valérie Gros[7], Joanna Kolasinski[11], Morgan Lopez[7], Olivier Magand[9], Erwan Noury[9], Manuel Nunes-Pinharanda[6], Guillaume Payen[9], Joris Pianezze[1], David Picard[4], Olivier Picard[2], Sandrine Prunier[2], François Rigaud-Louise[2], Michael Sicard[2], and Benjamin Torres[5]

[1]LAERO, Laboratoire d'Aérologie (UMR 5560 CNRS, UT3, IRD), Toulouse, France
[2]LACY, Laboratoire de l'Atmosphère et des Cyclones (UMR 8105 CNRS, Université de La Réunion, Météo-France), Saint-Denis de La Réunion, France
[3]Lab-STICC (UMR 6285, CNRS, ENSTA-Bretagne), Brest, France
[4]LAMP, Laboratoire de Météorologie Physique (UMR 6016 Université Clermont Auvergne, CNRS), Aubière, France
[5]LOA, Laboratoire d'Optique Atmosphérique (UMR 8518 Université de Lille, CNRS), Villeneuve d'Ascq, France
[6]LATMOS,Laboratoire Atmosphères, Observations Spatiales, (UMR 8190 IPSL, UVSQ, Université Paris-Saclay, Sorbonne Université, CNRS), Guyancourt, France
[7]LSCE, Laboratoire des Sciences du Climat et de l'Environnement (UMR 8212 IPSL, CEA-CNRS-UVSQ, Université Paris-Saclay), Gif-sur-Yvette, France
[8]MIO, Institut Méditerranéen d'Océanologie (UMR 235 Aix Marseille Univ, Université de Toulon, CNRS, IRD) Marseille, France
[9]OSU-R, Observatoire des Sciences de l'Univers de La Réunion (UAR 3365 CNRS, Université de La Réunion, Météo-France), Saint-Denis de La Réunion, France
[10]TAAF, Terres australes et antarctiques françaises, Saint-Pierre de La Réunion, France
[11]ENTROPIE, Écologie Marine Tropicale des Océans Pacifique et Indien (UMR 250 Université de La Réunion, IFREMER, CNRS, IRD, Université de Nouvelle Calédonie), 97744, Saint-Denis, La Réunion

**Correspondence:** Tulet Pierre (pierre.tulet@aero.obs-mip.fr)

**Abstract.** This article is devoted to the presentation of the MAP-IO observation program. This program, launched in early 2021, has enabled the observation of nearly 700 days of measurements over the Indian and Southern Ocean thanks to the equipment of 17 meteorological and oceanographic scientific instruments on board the ship $MarionDufresne$. Several observation techniques have been developed to respond to the difficulties of observations on board ships, in particular for passive remote

5   sensing data, as well as quasi-autonomous data acquisition and transfer. The first measurements made it possible to draw up unprecedented climatological data of the Southern Ocean of the size distribution and optical thickness of aerosols, of the concentration of trace gases and greenhouse gases, of UV, and of integrated water vapor. High resolution observations of phytoplankton in surface waters have also shown a great variability in latitude, in terms of abundance and community structure (diversity). The operational success of this program and these unique scientific results all together establish a proof of concept

10   and underline the need to transform this program into a permanent observatory.





# 1 Introduction

Due to its remoteness, the Southern Ocean (south of 35°S) is probably one of the least studied oceans in the world. Recently, Skinner et al. (2020), showed the important role of Southern Ocean convection as a potential amplifier of Antarctic warming and atmospheric $CO_2$ rise. The circumpolar current is the most powerful ($130\ \mathrm{Mm^{-3}\ s^{-1}}$) and has the strongest surface currents (0.9 to $3.7\ \mathrm{km\ h^{-1}}$) in the world. This strong oceanic current associated with quasi-permanent strong surface winds of the meridional overturning circulation create the conditions of important ocean-atmosphere exchanges that contribute significantly to the earth climate budget (e.g., Mayewski et al., 2009; Abernathey et al., 2011; Marshall and Speer, 2012; Nicholson et al., 2022). For example, Gruber et al. (2019) estimates that 40 to 50% of the global absorption of atmospheric $CO_2$ by the ocean occurs in the Southern Ocean. However, in the context of climate change, the evolution of the ocean $CO_2$ sink remains uncertain. This uncertainty is particularly strong in the Indian and Southern Ocean due to the lack of atmospheric observation to better constrain inversions (Le Quéré et al., 2007; Landschützer et al., 2015; DeVries et al., 2017) and the lack of seasonal and intra-seasonal observations. The SOCAT (https://socat.info, last access: 8 November 2023) and GLODAP (http://www.glodap.info, last access: 8 November 2023 (Lauvset et al., 2021)) databases illustrate the crucial lack of data collection of this region in comparison with those carried out of the North Atlantic or the equatorial Pacific.

From an atmospheric point of view, the lack of atmospheric observation over the Southern Ocean poses several problems as well for numerical weather forecasts (data assimilation), for climate models (long term observation), and for calibration/validation of spaceborne sensors. Recently, the WMO has emphasized this need to address operational and scientific issues (Thurston et al., 2021). For example, the observation of atmospheric water vapor is crucial because of its key role in the weather and climate system. Its spatio-temporal evolution is at the origin of many meteorological phenomena, sometimes intense and poorly modeled by numerical weather forecasting models. In the longer term, its evolution is an indicator of climate change through its strong link with atmospheric temperature. The world's oceans produce nearly 86% of atmospheric water vapor (Bengtsson, 2010) but are the areas where its observation is most patchy, being limited to surface or satellite observations (Smith et al., 2019). The spatial distribution of key radiatively active trace gases, such as ozone in the stratosphere, is largely affected by the Brewer-Dobson circulation (BDC) characterizing three latitudinal regions from the equator to the pole (Butchart, 2014): (i) the tropical stratosphere reservoir, (ii) the strong mixing mid-latitude surf zone and, (iii) the polar vortex. Those regions are separated by a permanent subtropical dynamical barrier and a winter polar barrier. Chemistry-climate and climate models predict a strengthening of the BDC, especially within its shallow branch, due to the increase of greenhouse gases (Abalos et al., 2021). Although two stations of the NDACC network (https://ndacc.larc.nasa.gov) are located in Reunion Island (21°S) and on the island of Kerguelen (49.3°S), the Indian ocean is almost absent from the operational monitoring networks of the atmosphere to monitoring changes in both barriers. The observation of ozone over a large region of the Indian Ocean with repeatable trajectories will allow a robust characterization in the different regions separated by those dynamical barriers. Within the troposphere, understanding the transport and the aging of aerosols over sea is an important challenge both for the climate budget and for the numerical weather forecast. The formation, the cloud condensation nuclei (CCN) and the ice forming nuclei (IFN) properties of marine aerosols are still poorly understood, especially in areas where the production of organic matter by





phytoplankton is important (Charlson et al., 1987; Quinn and Bates, 2011; Sellegri et al., 2021). The emission processes of marine aerosols and sea spray are also poorly known and parameterized in numerical models under strong wind and heavy swell conditions (Canepa and Builtjes, 2017; Pianezze et al., 2018; Sauvage et al., 2021). The challenges relate in particular to our ability to better understand and predict storms, deep convection, and tropical cyclones (Ramanathan et al., 2001; Hoarau et al., 2018; Sroka and Emanuel, 2021). Moreover, the Southern Indian Ocean, mainly loaded in sea salt aerosols, is also

impacted by long range transport pathways connecting South America, southern Africa, Australia, and South-East Asia to this part of the world, leading to a low yet highly variable aerosol burden (Duflot et al. (2022), and references therein). Indeed, these source regions are yearly submitted to the Southern Hemisphere biomass burning (BB) season, and show records of extreme wildfires (e.g. the 2020 Australian wildfires). These BB events emit large quantities of gases and particles into the atmosphere (Andreae and Merlet, 2001), such as carbon monoxide (CO) and fine smoke particles (black carbon (BC) and organic matter).

Chemistry in the fire plumes involving CO may lead to the formation of tropospheric and stratospheric ozone (Crutzen and Andreae, 1990), which may exert a significant climate forcing in downwind regions. Emitted BC particles are highly absorbing by nature and contribute to reducing the cooling produced by scattering-dominated carbonaceous aerosols (Jeong and Wang, 2010). This effect is highly season-dependent and can extend to greater scales (from regional to global) when the particles are injected in the stratosphere. The penetration of smoke-related compounds in the stratosphere is thought to be more frequent in

a warming world, and depends on pyroconvection mechanisms (Fromm et al., 2000) as well as on the tropopause which acts as a dynamical barrier. Therefore, the climatic impacts of such plumes have to be assessed in this poorly documented part of the world.

From the point of view of marine microbial ecology, the Southern Ocean is depicted as the most productive ocean on Earth; with a clear boundary between the South Western oligotrophic Indian Ocean dominated by cyanobacteria, the Subantarctic

Zone dominated by haptophytes ($< 10 \, \mu m$), and south of the Polar Front dominated by nano-eukaryotes (a polyphyletic group with cell size between 2-3 and 20 $\mu m$) and diatoms (Iida and Odate, 2014). The direct link between phytoplankton production and krill biomass sustaining the trophic conditions in the area makes phytoplankton a major group to study, especially in the current expected climate changing conditions. A large part of the export of carbon in the area is controlled by the mixing pump (Nowicki et al., 2022), one of the most affected directly by the temperature increase. Contrasted scenarios are expected, indeed,

air temperature increase in the South Indian Ocean is expected to amplify the oligotrophic conditions led by stratifications, and reversely, intensify wind blowing due to ENSO variability (Wang et al., 2022), the latter would increase mixing, enhancing primary production by favorizing the input of nutrients produced deeper by mineralization of the organic matter. Both scenarios will clearly influence the carbon biological uptake of $CO_2$ and the trophic status of the area. In Auger et al. (2022), the Southern Ocean below the South Western Indian ocean has evidenced a small decrease in surface temperature, suggesting an increase

in mixing with potential increase in production, but this does not account for the deep layers of global heating (Sallée, 2018). Little knowledge on distribution of phytoplankton functional groups in the Southern Ocean and South Western Indian ocean, as well as in the highly productive fronts, make the estimation of the biological pump and the potential of energy transfer to the higher trophic levels difficult to forecast. The direct relation observed between phytoplankton functional groups distribution and CCN (Sellegri et al., 2021) makes the investigation of both variables crucial for a holistic understanding of the effect of



climate change in the study area.

The MAP-IO ($Marion Dufresne$ Atmospheric Program - Indian Ocean) program aims to overcome the lack of observation in this region of the Earth, which is poorly documented compared to the other oceans, by equipping the $Marion Dufresne$ vessel (https://taaf.fr/en/marion-dufresne-and-astrolabe, last access: 8 November 2023) with a set of in-situ and remote sensing instruments for atmosphere and marine studies. This program has been labeled by the French Commission Nationale de la

Flotte Hauturière (CNFH, https://www.flotteoceanographique.fr, last access: 8 November 2023) for the period 2021 to 2024. During this period, MAP-IO will operate as a scientific program for the acquisition and scientific enhancement of four years of data. This period will also serve as an operational prototype to study the feasibility of switching the program to a permanent observatory aimed at integrating the international infrastructures networks such as ACTRIS (https://www.actris.eu, last access: 8 November 2023) or ICOS (https://www.icos-cp.eu, last access: 8 November 2023). This article aims to present the MAP-IO

program. It is organized as follows. The first part will present the objectives and the operating framework of the program. A second section will present the instrumental setup while a third one will describe the information technology deployed for this purpose. The acquisition and archiving of data as well as the website of the program will be presented in the third part. A preliminary presentation of the scientific results after 30 months of data collection will constitute the fourth part. The last section will be devoted to the conclusion and perspectives.

## 2   Framework and objectives

The $Marion Dufresne$ is a large multipurpose vessel (120 m long and 4900 t). Under charter by the TAAF (https://taaf.fr/en, last access: 8 November 2023), this vessel is intended for the supply and transport of personnel on the Southern lands and on the Scattered islands of the Mozambique Channel. It generally performs 4 annual rotations from Reunion Island (home port) to the islands of Crozet, Kerguelen, and Amsterdam (123 days a year). The rotation to the scattered islands is less frequent

(once every four years or so) and is done around Madagascar through the Mozambique Channel. The rest of the year (217 annual days on average) the ship is devoted to scientific research. It is then managed by the French oceanographic fleet (FOF, https://www.flotteoceanographique.fr/en) and is operated on various sea campaigns whose scientific proposals are evaluated by the CNFH.

The MAP-IO program is to carry out atmospheric observations at the ocean-atmosphere interface and integrated on the atmo-

spheric column over the entire globe with a focus of particular interest for the study of the Indian and Southern Oceans. The goal of the MAP-IO program is to study the feasibility to establish a permanent marine observatory onboard the $Marion Dufresne$. The program has three main objectives; (i) integration into international atmospheric and oceanic networks by providing high quality data on a region devoid of permanent observation, (ii) validating and calibrating space sensors and numerical weather forecasting models, and (iii) monitoring global changes and interannual variability by continuous observation of atmospheric

and phytoplankton over the Indian and Southern Oceans. MAP-IO takes advantage of the various scientific campaigns at sea planned by the FOF by exploring different poorly documented ocean areas and by increasing and completing the observation systems of the programs with additional measurements. The differentiation value of MAP-IO versus conventional scientific



programs at sea, is that it also relies on the regular rotations of the TAAFs dedicated to visit the French Austral islands by documenting the atmosphere and the ocean surface states over a wide variability of seas and latitude conditions. These multi-year observations of the Indian and Southern Oceans make it possible to uniquely document the trends, the mechanisms of ocean-atmosphere exchanges, and the atmospheric composition on a seasonal and interannual basis. The large collection of the in-situ data of MAP-IO under different latitudes and seasons, sea state, and meteorological conditions should provide important potentialities of machine learning uses. The potential near-real-time transmission of most MAP-IO observations is intended to fill an important data gap over the Southern Ocean in the assimilation or validation-calibration processes of numerical air quality or forecast models. Launched at the beginning of 2021, MAP-IO has already recorded in July 2023 nearly 700 measuring days at sea ($\sim$ 75 % of the time). In addition to the 9 TAAF's rotations on southern lands, the program's instruments have been previously utilized on 12 scientific campaigns dedicated to various fields such as volcanology, geology, geochemistry, sedimentology, oceanography, and marine biology.

## 3 Description of instruments and treatment methods

### 3.1 Instrument locations

The $Marion Dufresne$ Scientific vessel has embedded 17 scientific instruments on board, representing more than 25 atmospheric and marine biological parameters. Figure 1 illustrates the location of these instruments and computer servers on the ship. The optical measurements of cloudiness and UV radiation are located on the front mast of the ship at 20 m above sea level (a.s.l.). This forward position is a trade-off limiting the pollution of the measurements by the exhaust fumes of the ship. Under this mast is located the dedicated computer acquisition room. The gas and aerosol inlets are located on deck i (20 m a.s.l.), oriented towards the front of the deck to reduce the possible contamination of the air sampled by the ship's activities. The in-situ analysers are located below the inlets. All the instruments and data acquisition computers are mounted on a shock absorbing table in order to preserve the durability of the systems in particular during strong swell conditions. This room is located next to the wheelhouse that facilitates the maintenance of instruments. Note that for the quality of the atmospheric aerosol measurements, particular attention was paid by minimizing the distance (8 m) and the elbows of the air conveying hose pipes. After testing several GNSS (Global Navigation Satellite System) antenna locations on the ship, the antenna has been located at the back of the vessel on deck J. This particular location was constrained to limit signal interference with the ship's instrumentation (radar, iridium beacon). Meteorological stations, the sun/sky/moon photometer, and the mini-SAOZ are located on the ship handle of the vessel at 25 m a.s.l., at about 10 m in front of the funnel. An automated pulse shape recording and imaging flow cytometer (Cytobuoy, NL) is installed on a bench in a dry laboratory close to the thermo-salinograph (SeaBird, SBE21) which allows it to be coupled with the ship's clean seawater circuit. The last area concerns the concentrators servers located in the information technology (IT) room (Informatic room). These servers are connected to the IT system of the vessel. This connection allows us to be connected to the satellite internet network of the vessel and to get the ship location and some sea surface measurements such as SST, salinity, sea wave and currents. All instruments and data acquisition methods are described below and summarized in table 1.



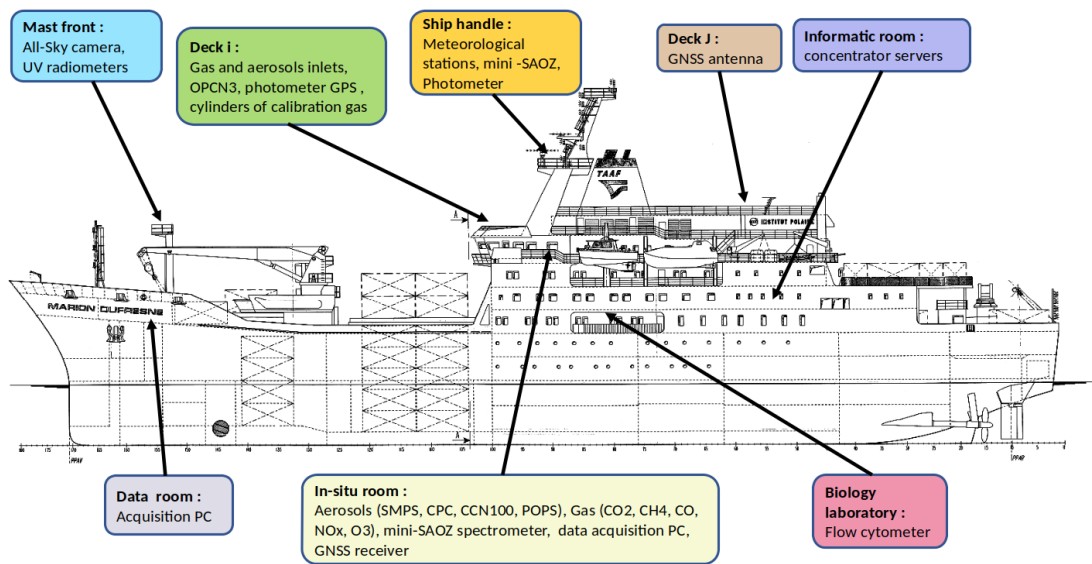

**Figure 1.** Location of the MAP-IO instruments and informatics servers onboard the $Marion Dufresne$ vessel.

## 3.2 Pulse shape recording flow cytometry

Pulse shape recording and imaging automated flow cytometer such as the Cytosense (b.v.) has been used successfully on ships of opportunity, scientific vessels, a buoy, and in coastal observatories for several weeks without human action, collecting data on phytoplankton size classes at the single cell level in an autonomous way (Thyssen et al., 2008, 2011, 2015; Marrec et al., 2018; Louchart et al., 2020). Phytoplankton abundance and functional groups are resolved on the basis of their size and pigment content when cells pass in front of a laser beam. The Cytosense automatically analyzes samples for phytoplankton counts in the size range of 0.6-800 μm in width and several mm in length. The sea water sample is funneled with a weight calibrated sample pump in an injector where it is surrounded by an isotonic sheath fluid, generating a laminar flow forcing particles in a single-file, before entering the measurement cuvette where it goes through a 120 mW 488 nm laser beam (Coherent®). Doing so, a set of optical curves, called pulse shapes, are generated for each recorded particle. The pulse shapes of side-ward scatter (SWS, 488 nm) and fluorescence emissions were separated by a set of optical filters (orange fluorescence (FLO, 515–650 nm) and red fluorescence (FLR, 668-726 nm) and collected in photomultiplier tubes. The pulse shapes of forward scatter (FWS) were collected on left and right-angle photodiodes and used to validate the laser alignment. Samples are scheduled to be analyzed every two hours from a stabilized $300 \, \text{cm}^{-3}$ sub-sampling chamber before the acquisition. The instrument and the acquisition protocol are described in (Marrec et al., 2018). For the identification of phytoplankton groups, two protocols were successively run. One triggering on FLR 5 mV for 5 min targeting RedPicoProk and OraPicoProk and a second one triggering on FLR 20 mV for 10 min targeting the RedPico, RedNano, OraNano, HsNano, RedMicro, and OraMicro phytoplankton





groups (Thyssen et al., 2022). Phytoplankton groups were manually classified using the CytoClus® software by generating several two-dimensional cytograms plotting descriptors of the four pulse shapes such as the area under the curve of the pulse-shape signals (Total FWS). Group abundance and cell properties were processed by the software. The size of the different phytoplankton cells was estimated based on the relationship between silica beads real sizes (1.0, 2.01, 3.13, 5.02, 7.27 μm non-functionalized silica microspheres, Bangs Laboratories, Inc.) and Total FWS signal and then converted into equivalent spherical diameter (ESD) and biovolume. A power-law relationship following the procedure in Marrec et al. (2018) allowed the conversion of the Total FWS signal into cell size. The stability of the optical unit is routinely checked thanks to a dedicated filled syringe with a solution of diluted 2 μm red Polyscience, inc., polystyrene fluorospheres. A CCD camera installed in front of the measuring cuvette collects images of cells that were pre-defined in a FWS versus FLR cytogram. The resolution of 3.6 μm per pixel allows the identification of phytoplankton cells above 10 μm at a genus level.

### 3.3    In-situ atmospheric measurement

The inlets of the aerosols and gases instruments are located on deck "i". Aerosol analysers are installed downstream of a dedicated inlet equipped with a Nafion dryer, and temperature and water vapour sensors. A dispatcher distributes the sampled flow to instruments. All instruments' outputs are filtered to delete data potentially polluted by the ship's exhaust smoke. Two methods are used:

- A dynamic approach is based on the relative wind direction and intensity. The data is filtered when the air inlet is downwind of the chimney or when the wind is weak that it permits contamination by turbulent diffusion. Sensitivity studies have set a wind direction at 145° associated with a cone angle of 20° and a threshold on the wind speed at 2 $\mathrm{m\ s^{-1}}$.

- A chemical approach is based on CO and $\mathrm{NO_x}$ measurements. We assume that a short and high increase of CO or $\mathrm{NO_x}$ cannot occur in marine environments and comes from a local combustion process. A concentration peak filtering approach developed by the ICOS IR is added to the dynamic method. The spike detection algorithm is described by El Yazidi et al. (2018). The advantage of this second approach is that it makes it possible to filter the pollution linked to the ship's activity which would not necessarily come from exhaust smoke.
The other sources of pollution are filtered manually by the PIs of the instruments.

#### 3.3.1    Greenhouse gases

A complete equipment for greenhouse gas (GHG) measurements has been installed onboard the ship. The setup includes an air inlet connected to a continuous high precision analyzer through a 1/4" Dekabon tubing. The analyzer (Picarro G2401 CFKADS-2372) provides $CO_2$, $CH_4$, CO, and $H_2O$ continuous measurements. 2 μm and 0.5 μm filters are placed in the inlet line to avoid large particles entering the analyzer. In addition, a Nafion membrane is placed at the instrument inlet to dry ambient air prior injection into the analyzer. A set of 5 compressed air cylinders, initially calibrated at LSCE, (reference scales WMO-CO2-X2019, WMO-CH4-X2004A, WMO-CO-X2014A) and associated pressure regulators are used for calibration and



quality control. The air inlet, calibrations and quality control cylinders are connected to a multi-position valve (Valco valve) that is controlled by the GHG instrument through predefined automated measurement sequences. The analyzer is calibrated once a month with 4 cylinders. The fifth cylinder is used as a target gas which is measured daily for 30 minutes in order to estimate the repeatability and precision of the measurements. Data are automatically transferred to the SNO ICOS France database and are automatically treated in near real time using prescribed algorithms (Hazan et al. 2016). Final quality control is made by the station PI.

### 3.3.2 $O_3$ and $NO_x$

The Model Teledyne N500 CAPS $NO_x$ analyzer uses superior Cavity Attenuated Phase Shift (CAPS) Spectroscopy to measure "true" $NO_2$, $NO_x$, and NO gases. The instrument combines direct $NO_2$ measurements with highly efficient gas phase titration (GPT) to convert and measure the NO gas component. An automatic baseline reference cycle accounts and compensates for any potential baseline drift due to varying environmental conditions. Detection limit is 0.1 ppb. The model HORIBA APOA-370 analyzer continuously monitors atmospheric ozone concentrations using a cross flow modulated ultraviolet absorption method. It uses heated de-ozonizer to remove any $O_3$ in the reference gas to reduce interference, eliminate moisture interference. The setup includes an air inlet connected to a manifold through a 1/4" PTFE tubing. $NO_2$, NO and $O_3$ are monitored every minute. A model 146i Multi-gas Calibrator is used to calibrate ozone and $NO_x$ analyzer at a monthly frequency.

### 3.3.3 Aerosol total number concentration

We use a water-based condensation particle counter (CPC, model MAGIC) able to measure the total number of aerosols with diameter ranging 5 nm to 2.5 μm. A water-based rather than butanol-based CPC was chosen due to safety considerations. The upper limit of concentrations being detected is $4 \cdot 10^5 \text{cm}^{-3}$.

### 3.3.4 Aerosol size distribution

Submicron size distributions are measured with a Scanning Mobility Particle Sizer (SMPS) (model 4S). The instrument is composed of a Differential Mobility Analyzer (DMA) coupled with a CPC (model MAGIC) that provides the number size distribution of aerosols from 10 to 350 nm. Scanning mobilities is performed in an up-scan and down-scan mode alternatively. Quality-control of the instrument is made by comparing total number concentrations calculated from the SMPS with those of the total CPC count. The OPC-N3 optical particle counter was chosen to be installed on the deck to measure the aerosol size distribution of the largest particles. OPC-N3 can measure the aerosol concentration and size distribution from 350 nm to 40 μm. The precision of an OPC-N3 is unknown but three instruments were deployed simultaneously to estimate the inter-instrument uncertainty.





### 3.3.5 CCN

Cloud condensation nuclei chamber (model DMP CCN-100) counts and sizes of aerosols that can be activated into cloud
225 droplets. Aerosols enter into a thermal-gradient diffusion chamber where a supersaturated water vapor condition is created by
the difference in diffusion rates between water vapor and heat. Then the cloud droplets formed are counted and sized using
an optical light-scattering counter (range 750 nm to 10 μm). The CCN-100 scans the CCN aerosols properties at various
supersaturations ranging from 0.07% to 2%.

When the SMPS was under maintenance (between April and June 2021 and during the year 2022), it was decided to shut down
230 the other instruments. After filtering, the in-situ aerosol instruments operated on average 50% of the time, representing 33891
data for the SMPS and 24680 data for the OPC-N3.

### 3.4 Passive remote sensing

### 3.4.1 Aerosol Extinction Optical Depth (AOD) and water vapor

The shipborne CIMEL CE318-T was developed in the frame of the AGORA-Lab program (https://www.agora-lab.fr, last
235 access: 8 November 2023) to enable spectral AOD (340 to 1640 nm), spectral downward atmospheric radiances (440 to
1020 nm), and water vapor measurements on mobile platforms and to expand the AErosol RObotic NETwork (AERONET)
coverage over the vast ocean area. It also intends to cover the need for future mobile exploratory platforms, like those scheduled
in ACTRIS infrastructure. A prototype version of this ship-borne photometer, co-located with lidar and a microtops hand-held
photometer, was set up and operated successfully during two R.V. Polarstern trans-Atlantic cruises in 2018 (Yin et al., 2019)
and in a similar way during the TRANSAMA campaign between Reunion Island and Barbados in April/May 2023.

The system is composed of an optical head, a rotational base, its control unit, air-pumping unit, weather stop unit, GPS-based
compass, and positioning system units (date, time, geo-location, heading, pitch, and roll). The optical head is the standard
CE318-T with version 1 of the ship-borne software. The GPS receivers were fixed on the platform together with the photometer
robot. In order to track the sun and the moon continuously, the system first targets the sun/moon with the last received time,
geo-location, heading, pitch, and roll information. When the sun/moon enters into the tracking system's field-of-view, the
photometer switches into tracking mode like a regular AERONET instrument.

Before the instrument was permanently set up on the $Marion Dufresne$ several test campaigns such as AQABA, around
the Arabian peninsula in 2017 (Unga et al., 2019), and OCEANET transatlantic campaigns with the R.V Polarstern (Yin
et al., 2019) have been necessary to converge to an operational mobile solution for the harsh marine environment. The air-
pumping unit creates cleaned and compressed air to the collimator to prevent contamination of the optics by sea spray. The
standard CE318-T resistive wet sensor was replaced by a more appropriate optical rain sensor to prevent the degradation due
to the strong corrosion. An anemometer has been added to stop the operation when wind speed is too high and may produce
problematic vibrations. The estimated impact of the smoke plume emitted by chimneys is quite negligible when compared to
AOD uncertainty. The post-field calibration has been performed and confirms the negligible calibration coefficient changes
during the 14 months of continuous operation in this harsh environment.



As it is based on the standard CE318-T version, the shipborne sunphotometer is fully compatible with AERONET calibration and the quality control/quality assurance (QC/QA) procedures. As the instrument is AERONET compatible, this yields huge simplification in processing. Once raw data are collected, they are transmitted via satellite to the server of the PHOTONS CNRS National Observation Service (University of Lille) at day+1, for near real time processing of the spectral AOD, water vapor
content, Angström exponent, and downward atmospheric radiances in the AERONET version 3 processing system (Giles et al., 2019). In addition to AERONET processing and archiving, a near real time visualization system has been developed (https://aeronet.gsfc.nasa.gov, last access: 8 november 2023). All the data are transferred and available at the french national AERIS database in near real time, as for any AERONET station.

### 3.4.2 Integrated $O_3$, $NO_2$ and $H_2O$

A Mini-SAOZ is the modernized and lightened version of the SAOZ (Système d'Analyse par Observation Zénithale) instrument developed at the end of the 80s by Pommereau and Goutail (1988). The instrument is completely automatic and self calibrated. It was installed on the $Marion Dufresne$ in January 2021 after being adapted for mobile and marine conditions. The Mini-SAOZ was successfully compared to other NDACC instruments during the CINDI 2 campaign at Cabauw, the Netherlands (Kreher et al., 2020). The Mini-SAOZ uses a miniaturized Czerny-Turner spectrometer equipped with a flat-field grating and
a two-dimensional CCD detector of 2048 × 16 pixels. The entrance slit and the grating are adapted to allow an average resolution of the order of 0.7 nm in the range 300-800 nm. The light passes through the optical head linked to a 30 m optical fiber which brings the light to the spectrometer located inside the ship, in the weather room. The instrument's field of view is 8°. A GPS with its marine antenna is installed next to the optical head, allowing accurate time and the location of the measurement. The instrument is coupled to a robust on-board computer with specific software that controls the instrument,
acquisition, spectral analysis, and data storage. The following three steps summarize the Mini-SAOZ processing chain from acquisition to measurement of the vertical column:

- Spectra measurements are taken from sunrise to sunset up to a solar zenith angle of 96°. The exposure time is automatically adjusted and the spectra are added to memory in a 60 second cycle. GPS time and location data are used for the accurate calculation of the Solar Zenith Angle (SZA) of each measurement. The dark current is measured using a
mechanical shutter and stored. It is subtracted from each measured spectrum and after applying calibration data. The corrected spectra and other parameters such as the GPS location and the temperature inside the instrument are saved in a corresponding binary file or level 0;

- Conversion of slant columns into total columns (level 2 data) using an AMF following the recommendations of the WG-UVVIS NDACC. For ozone AMF daily values are calculated by the UVSPEC/DISORT radiative transfer model
(Mayer and Kylling, 2005). The model uses a multi-input TOMS version 8 (TV8) database of climatological ozone and temperature profiles (McPeters et al., 2008). See more details in Hendrick et al. (2011). For $NO_2$ the process is similar to that of ozone but the AMFs are different for morning and evening twilight in order to take into account the diurnal variation of the constituent. For $H_2O$, AMFs from a Sarkissian model (Sarkissian et al., 2009) are used;



- The spectral analysis is carried out by the computer on board the $Marion Dufresne$ allowing the calculation of the SCD of ozone; $NO_2$ and $H_2O$ in real time. Then, the conversion into a vertical column is carried out after receipt of the level 1 data in near real time (generally d+2) by a centralized data system at the LATMOS laboratory in Guyancourt, France. The real time processing was updated to a mobile platformrequiring a fairly large calculation time. New filters were developed to avoid the impact of single saturated spectrum during the integration time cycle. Level 2 data is then made available on the SAOZ page: http://saoz.obs.uvsq.fr/MarionDuf.html (last access: 8 November 2023).

### 3.4.3 GNSS Integrated water vapor

The opportunity measurement provided by the GNSS which allows the restitution of integrated water vapor content (IWV). This restitution is common for fixed terrestrial GNSS antennas (Bosser and Bock, 2021) but is original for a ship-borne GNSS antenna, all the more so in the operational context of an atmospheric observatory. The complete methodology for GNSS IWV retrieval is described in Bosser et al. (2022). The analysis of raw GNSS phase measurements is performed in PPP (Precise Point Positioning) mode. This mode of analysis does not require reference ground stations, which is particularly suitable in the marine context where the antenna is potentially very far from the coast. At the end of this analysis, the estimated positions and the Zenith Total Delay (ZTD) are available at a rate of 30 or 300 s (depending on the type of analysis). ZTDs are then converted into IWVs from surface measurements made by the onboard weather station. Two routine analyses of GNSS data are in place. Only GPS data are currently considered:

- Ultra analysis: this analysis uses the so-called "ultra-rapid" products made available by Jet Propulsion Laboratory (JPL), which provide, among other things, the precise orbits of the satellites in the Global Positioning System (GPS) constellation as well as the corrections to their atomic clocks. As these products are available on day+1, this analysis is carried out daily on day+1. The temporal resolution of the solution is 300 seconds (imposed by the temporal resolution of the orbits and clocks).

- Rapid analysis: this analysis uses the so-called "rapid" products made available by JPL which provide, among other things, the precise orbits of the satellites in the GPS constellation as well as the corrections to their atomic clocks, which are more accurate than the ultra-rapid products. Since these products are only available on day+3, this analysis is performed daily from day+3. The temporal resolution of the solution is 30 s (imposed by the temporal resolution of the orbits and clocks).

More details could be found on the website: https://gipsy-oasis.jpl.nasa.gov/index.php?page=data (last access: 8 November 2023).

### 3.4.4 UV solar radiation

Three Kipp & Zonen broadband radiometers, SUV-A, SUV-B, and SUV-E have been deployed to measure ultraviolet irradiance across various spectral ranges at a frequency of 1 minute.



The SUV-A radiometer is specifically designed to measure UV-A radiation within the 315-400 nm range, with a yearly measurement drift of 5% and less than 1% non-linearity. The SUV-B radiometer is tailored for UV-B radiation measurement in the 280-315 nm range, also featuring a yearly 5% measurement drift and less than 1% non-linearity. Lastly, the SUV-E radiometer, whose spectral response aligns closely with the erythemal action spectrum (ISO 17166:1999/CIE S 007/E-1998), is similar to its predecessor, the UVS-E-T predecessor. It has a daily uncertainty under 5%, an annual sensor drift less than 5%, a directional response error under $5 \text{ m}^2 \text{ W}^{-1}$, and a non-linear error under 1%. SUV radiometers feature a photodiode detector, an optical filter, a diffuser, and a protective glass or quartz dome. The detection system integrates a photodiode, sensitive to UV radiation, and an optical filter, which defines the spectral response. The generated signal is amplified, and the output voltage, in combination with the instrument's sensitivity, is converted from volts to watts per square meter ($\text{W m}^2$). The white diffuser, positioned above the photodiode, ensures accurate directional response, while the protective dome safeguards it against debris and precipitation. The manufacturer offers a calibration accounting for the solar zenith angle and the total ozone column. Between each vessel rotation, the radiometers undergo recalibration at the Saint-Denis UV station, a part of the UV-Indien network (Lamy et al., 2021a), in Reunion Island. This process involves comparing the radiometers' readings with the calibrated values of a Bentham DTMc300 spectroradiometer (https://www.bentham.co.uk/products/components/dtmc300-double-monochromator-39, last access: 8 November 2023). By initially considering the manufacturer's calibration, relative differences between the radiometers and the Bentham measurements are identified and grouped by solar zenith angle (SZA) bands (approximately $\pm 5°$). A calibration coefficient dependent on the SZA is obtained by averaging these relative difference bins. Furthermore, the mean of all relative differences, irrespective of SZA, is computed to derive an overall calibration coefficient. This two-step calibration procedure compensates for instrument drift and variations in solar zenith angle and total ozone column.

### 3.4.5 Cloud nebulosity

The Reuniwatt SkyCam Vision is an all-sky camera designed to capture panoramic images of the sky in the visible spectrum every minute. Equipped with a $2048 \times 2048$ pixels CMOS sensor, its standard acquisition frequency is set to 30 seconds although it can be adjusted as needed. The camera utilizes the ELIFAN algorithm to calculate the cloud fraction (Lothon et al., 2019). This algorithm evaluates the R-B ratio distribution as well as each pixel's specific R-B ratio, applying various criteria and thresholds. The processing of the image involves several steps including defining the image contour, applying masks to the sun and other objects, identifying clear-sky and completely overcast images, and distinguishing between clear-sky and fully clouded conditions using either absolute or differential threshold methods.

## 4 The MAPIO IT architecture

The IT architecture is summarized in Figure 2. The challenge of onboard computing was solved by a judicious choice of robust computers with no moving parts and cases designed for natural cooling without fans, capable of withstanding shocks, vibrations, and sudden temperature variations.





**Table 1.** List of the MAP-IO instruments and specifications

| Instrument models - manufacturer | Type of measurements | Description |
|---|---|---|
| CytoBuoy Cytosense | Phytoplankton single cell | Flow cytometer including a 488 nm 120 mW primary laser. Camera module for imaging. Single particle optical optical pulse shape records. |
| Picarro G2401 | Greenhouse gas: $CH_4$, $CO_2$, $CO$, $H_2O$ | Near-infrared Cavity Ring-Down Spectroscopy. Repeatability at 5 seconds. |
| HORIBA APOA 370 | $O_3$ | Ozone analyzer using a cross flow modulated ultraviolet absorption method. Detection limit : 100ppt. |
| Teledyne API N500 | $NO_x$ | $NO_x$ analyzer combined with a "true" measurement of $NO_2$ with a CAPS and an efficient NO converter. Detection limit: 100 ppt. |
| TEI 146 i | Multi-gas calibrator | Calibrator for precision trace-level gas analyzers. |
| 4S SMPS | Aerosol size distribution | Sizing nanoparticles using differential mobility analysis. Measurement size range: 10 nm to 350 nm. Associated with CPC MAGIC. |
| Alphasense OPC-N3 | Aerosol size distribution | Laser optical diffusion technique. Measurement size range: 350 nm to 40 µm. |
| Aerosol Device Magic CPC | Aerosol number concentration | Aerosol counter using Milli Q water as growth fluid. Measurement size range: 5 nm to 2.5 µm. |
| DMT CCN-100 | Cloud condensation nuclei counter | Count and size of individual aerosol particles that can form into cloud droplets. Supersaturation generation (range 0.07 to 2%) and particle light-scattering counter (750 nm to 10 µm). |
| Kipp & Zonen UV radiometers | UV solar radiation | Three radiometers for solar ultraviolet radiation. Spectrums UV-A (320 nm to 400 nm), UV-B (280 nm to 315 nm), UV-E (erythemal irradiance). |
| Reuniwatt Sky Cam vision | Sky nebulosity | All-sky visible camera (type AV Mako G-419C) computing the cloud fraction every 5 min. |



| Cimel CE318-TS9 | Spectral AOD, atmospheric radiance and derived aerosol properties. Water vapor. | Automatic sun/sky/moon photometer. Updated for mobile observation and cold temperature. Spectral range 340 to 1640 nm. |
|---|---|---|
| Gordien Strato mini-SAOZ | $O_3$ - $NO_2$ - $H_2O$ integrated column | Automatic spectrometer. Updated for mobile observation. Spectral range 400 to 800 nm. |
| Vaisala WXT530 | Wind, T, Hu, Pressure, Rain | Two meteorological stations: ultrasonic wind sensors, capacitive sensor for barometric pressure, sensor for humidity and resistive platinum sensor for air temperature, acoustic sensor to measure precipitation. |
| Trimble Alloy GNSS receiver and Zephyr 3 Rugged antenna | IWV | GNSS receivers and antenna: acquisition of GNSS raw phase measurements for GPS, Glonass and Galileo constellations with a time resolution of 15 s. |

## 4.1 Data acquisition and transfer

Each instrument is connected to an acquisition computer which receives the measurements in real time and archives them for a period longer than a measurement campaign (i.e. for at least 3 months). As soon as a measurement is received on an acquisition
computer, it is automatically transmitted and stored on the two on-board concentrator servers installed in the "IT-room". Each concentrator server has 2 TB disks to archive data which represents two years of MAP-IO data storage. Automatic scripts are used to regularly delete old data on the acquisition computers to avoid disk saturation. The data acquisition scripts, installed on the acquisition computers, have been designed as real services that restart automatically whatever the cause of a possible stop. For the most sensitive services, it is possible to remotely take control of the computers electric boot relay cards using a
SSH command. The internet network of the vessel is a VSAT satellite link. This VSAT connection allows us to transfer the lightest data ($\sim$ 113 Mo per day) from the concentrator servers to the MAP-IO internet servers. While most of the measured data, after compression, can be transmitted within this constraint, the heaviest data such as the All Sky camera images or the cytometer data files cannot be transmitted daily. These data are then stored on both the data acquisition computers and the on-board concentrator servers for the duration of each campaign. At each stopover, all data stored on the concentrator servers
are then collected manually by the MAPIO staff and archived on the OSUR servers of the University of La Réunion. The data are therefore present on the acquisition PCs, on the two on-board concentrators and on the project's two internet servers.

## 4.2 Data access services

All data and services (FTP, HTTP, SQL) offered to the end user are within a secure area (DMZ) of the University of Reunion and maintained by the OSU-R (https://www.osureunion.fr, last access: 8 November 2023). Data access for users is provided

through secured FTP or secured MySQL as a pulling mode. Also, OSU-R pushes data to archiving centers such as AERIS (https://www.aeris-data.fr, last access: 8 November 2023) after formatting. OSU-R is in charge of long term data archiving.

### 4.2.1  Website service

A website service is available via a homepage with the URL: http://www.mapio.re (last access: 8 November 2023). The MAPIO website offered as open access:

- to monitor in real time the operating status of each instrument and data since the beginning of the campaign;

- to visualize by interactive graphics the data of each instrument in real time and in a geolocalized manner;

- to get information about all campaigns made and contacts of PIs and the technical staff.

The MAPIO website offered as restricted access:

- to consult the documentation and procedures related to each instrument (restricted access to MAP-IO participants);

- FTP access to all data of the program.

### 4.2.2  Monitoring service

The monitoring service is the main tool to assist in the diagnosis and maintenance of the IT systems and observing instruments. It is open to all data users and accessible by the MAP-IO website. At each port of call, a router equipped with a 4G SIM card is connected to the network of the acquisition computers. This allows the various PIs to make the necessary adjustments to the

instruments remotely. Due to the limited VSAT bandwidth, this possibility is not yet offered to users while the ship is on route.

### 4.2.3  Advanced data calculation service

Due to the pollution emitted by the ship's stacks, the in-situ measured data may not be usable in certain relative wind conditions. Therefore, an automatic flags calculation system has been set up on the project's web servers, indicating for each measurement whether it is likely to have been polluted by stack emissions.

The IWV data are calculated from the GPS Alloy data installed on the ship. These data are calculated by ENSTA Bretagne and integrated daily into the project's internet servers. They are therefore visualisable via the geolocated graph of the MAP-IO website.

### 4.2.4  Data transfer service

The web servers of MAP-IO provide users with secure access to data via the FTP protocol. This data can be both:

- daily data in the form of files sorted by date;

- files containing all the data for a campaign.



These latter files are only available at the end of each campaign. All data retrieved from the project's web servers are also archived in real time in a MySQL database. This archiving allows for easier and more accurate access to the data than using files obtained via FTP. PIs are responsible to analyze and validate data upon quality protocols defined by international standards.

Within one year, all the data acquired will have been validated by the PIs and transferred to international data centers such as AERIS (https://en.aeris-data.fr, last access: 8 November 2023) or to specific centers such as AERONET (https://aeronet.gsfc.nasa.gov, last access: 8 November 2023) or NDACC (https://www.ndaccdemo.org, last access: 8 November 2023).

## 5 Overview of the first results

### 5.1 Surface phytoplankton distribution

Phytoplankton cells were separated into 10 functional groups identified following the standard vocabulary (Thyssen et al., 2022); namely, RedPicoProk, OraPicoProk, RedPico, HsNano, OraNano, RedNano, OraMicro, and RedMicro. Samples were collected at a spatial resolution of $14 \pm 9$ km, sampled volumes averaged $1.04 \pm 0.36$ cm$^{-3}$ for FLR5, resolving the smallest sized groups (OraPicoProk, RedPicoProk), and $5.58 \pm 1.44$ cm$^{-3}$ for FLR20, resolving the other cited groups. Quality control was checked using the referenced 2 μm polystyrene red fluorescing beads and a maximal of 4% deviation around the Mean

of Total FWS was observed. RedPicoProk and OraPicoProk are the most abundant cells counted with median values of 1730 cells cm$^{-3}$ and 5130 cells cm$^{-3}$ with maximal values of 66080 on 21/07/2021 and 161900 cells cm$^{-3}$ on 19/05/2022 respectively (Fig. 3). Sizes estimated are the smallest within the phytoplankton cells observed with the flow cytometer with median values of 0.6 and 0.87 μm (Fig. 4) but maximal values reached 1.17 and 1.64 μm respectively. RedPico median abundance was of 3390 cells cm$^{-3}$ and the maximal value found was 27250 cells cm$^{-3}$ on 5 May, 2022 close to the South West African Coast

with a median size of 2.33 μm. The other groups' median abundance is below 500 cells cm$^{-3}$ but with inverse ESD (Fig. 4 and 5). As an example of a finding, there is a significant difference in the RedPico ESD between night and day samples, with larger cells during the day than during the night ($2.25 \pm 0.31$ and $2.37 \pm 0.30$ μm, $p < 0.001$), suggesting the influence of a diel cycle with a majority of divided smaller cells during the night. A significant difference in size between cells north and south of -40°S ($2.28 \pm 0.30$ and $2.43 \pm 0.40$ μm, $p < 0.001$) also evidence a difference in adaptation to warm subtropical waters and

cold subantarctic waters. Average smallest RedPico cells were found during April-May 2022 in the Mozambique Canal ($2.15 \pm 0.29$ μm) and during June and July north and north east of Madagascar island. A similar significant trend in night and day or north and south of -35°S in ESD differences is observed for OraPicoProk and RedPicoProk while RedNano do not evidence diel nor north south differences. The integrated camera collects photographs of the cells with a resolution good enough for the identification at a high taxonomic level for cells above 10 μm (Fig. 6). All cells are not pictured but enough per group of similar

pulse shapes. The collected photographs quantify the abundance of the most dominant taxonomic groups observed within the volume sampled. This volume sampled, although taken in less abundant surface waters, still remains enough to picture the diversity of the phytoplankton in southern areas or close to the islands.





**Figure 2.** IT architecture of the MAP-IO program: from acquisition to final users.

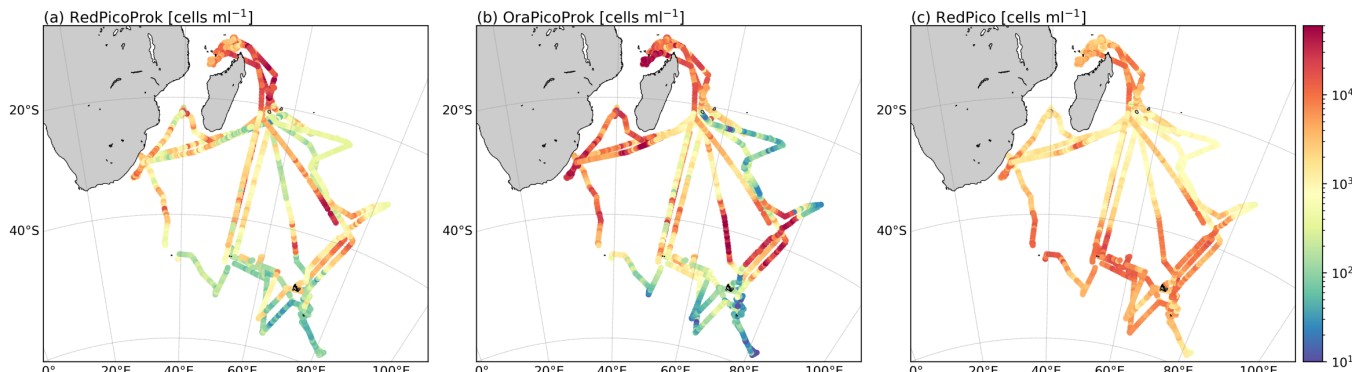

**Figure 3.** Abundance distributions for samples collected during the 2021-2022 MAP-IO cruises where the Cytosense instrument was in use for a. RedPicoProk (cells $cm^{-3}$) b. OraPicoProk (cells $cm^{-3}$) and c. RedPico (cells $cm^{-3}$). All abundances are available at https: //www.seanoe.org/data/00783/89505/ (last access: 8 November 2023). To limit the superposition of the trajectories, they were slightly offset.

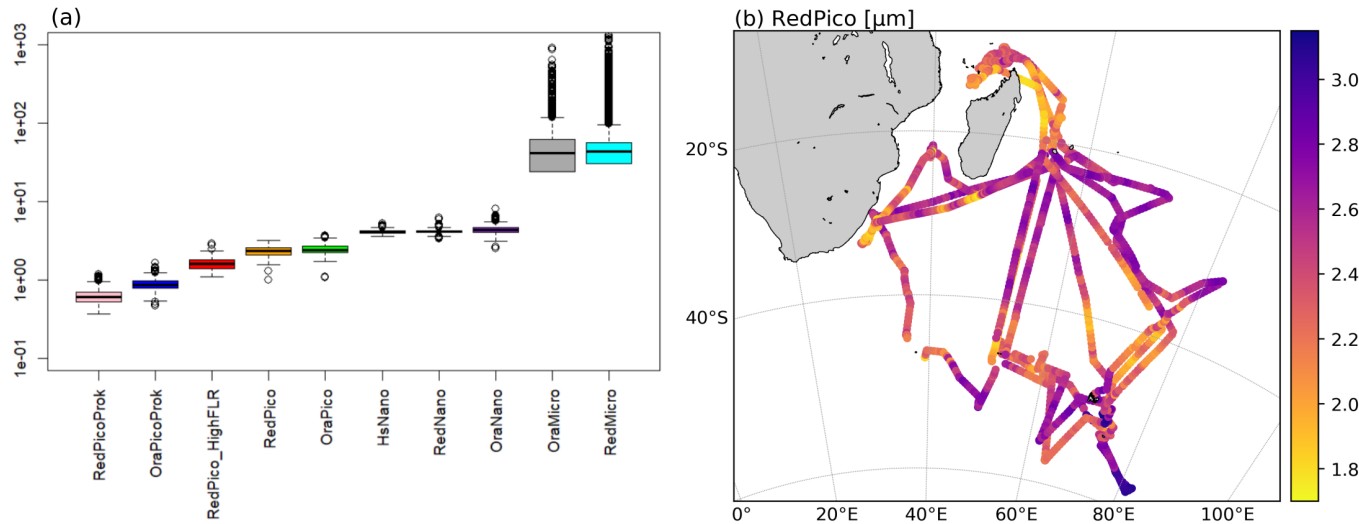

**Figure 4.** (a) Boxplot of the estimated ESD (μm) for all phytoplankton groups identified, except for OraMicro and RedMicro where length (μm) is used as they may correspond to chains of cells, from left to right: RedPicoProk, OraPicoProk, RedPico_HighFLR, RedPico, OraPico, HsNano, RedNano, OraNano, OraMicro, RedMicro. (b) Average ESD (μm) distribution for RedPico cells.

## 5.2 Gas distribution

The time series of $CO_2$, $CH_4$, and CO measurements carried out on board the $Marion Dufresne$ from October 2020 to
February 2023 are shown in Figure 7, with measurements from Amsterdam Island as the reference. Unsurprisingly, measurements are highly variable when the ship is in a harbour. There was also a high variability of measurements in the open sea

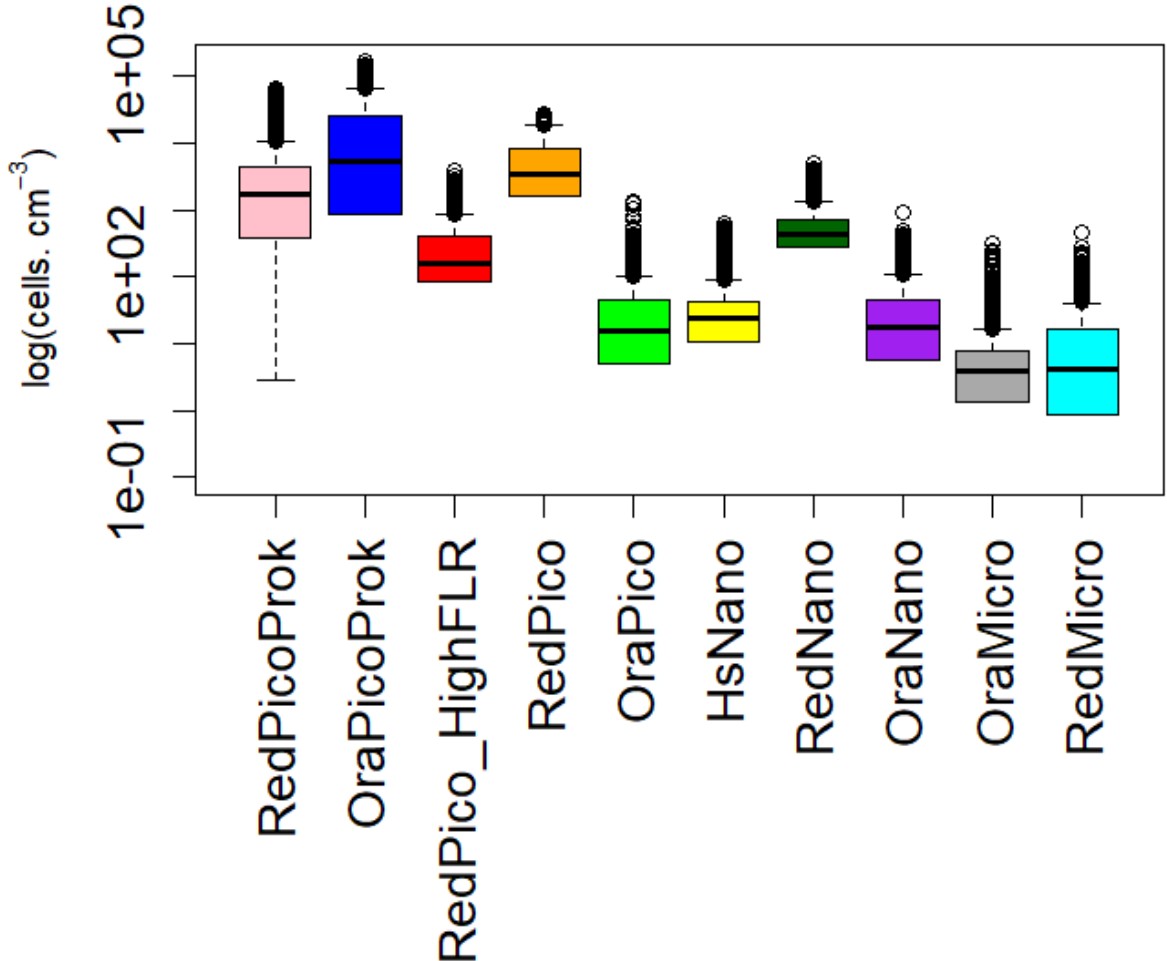

**Figure 5.** Boxplot of the abundance of the phytoplankton groups identified (cells cm$^{-3}$).

during the RESILIENCE campaign in March/April 2022, which can be explained by the fact that most of the campaign took place in areas fairly close to the coasts of Madagascar and South Africa. On the other hand, the passage close to the TAAF districts had only a very moderate impact on the concentrations of the three gases measured. In these measurement series, episodes of contamination by the ship's chimney were removed, based on the detection of CO peak concentrations (El Yazidi et al., 2018) and NO spikes (NO > 1 ppb). In total, the NO method leads to the filtering of 15.7% of the minute averaged measurements due to local pollution, while the use of CO leads to the elimination of 9.2% of measurements, about 40% of which




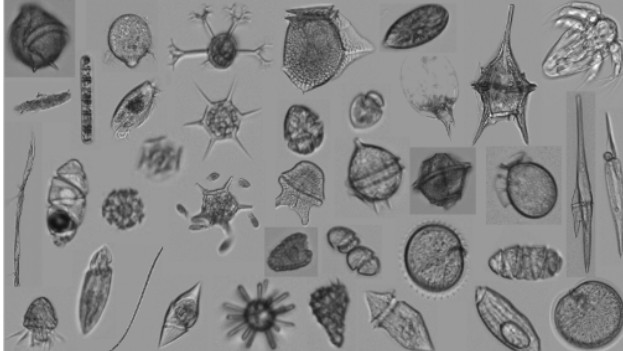
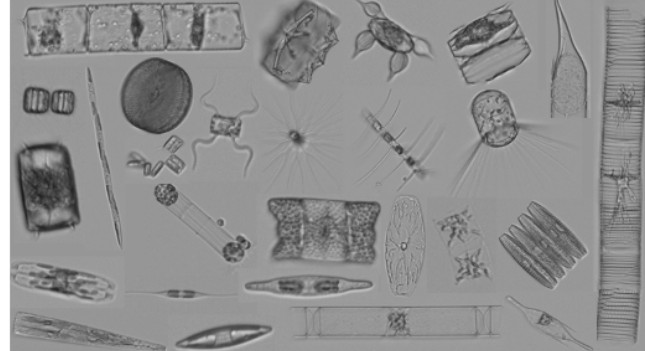

**Figure 6.** Random collection of phytoplankton pictures from the Cytosense image in flow device with a resolution of 3.6 pixels $\mu^{-1}$.

are shared with the NO method. Several high concentrations of $CO_2$ were thus suppressed, particularly on the transect between Crozet and Kerguelen. The higher frequency of contamination from the ship's stack in this area corresponds to a stronger tail-

wind. When the $Marion Dufresne$ passes close to Amsterdam Island (less than 5 km), measurements at both sites can be compared. Such comparisons are important to ensure the consistency of the Indian Ocean observation network (Amsterdam, Reunion Island, MAP-IO). The quality control carried out at each site with target gases does not enable the entire measurement chain to be controlled (e.g. possible leak on the sampling line, water vapour correction, etc.). Initial comparisons between the $Marion Dufresne$ and Amsterdam Island showed good correlation of $CO_2$ and $CH_4$ measurements, with differences of less

than 0.1 ppm and 0.5 ppb respectively for $CO_2$ and $CH_4$. However, a systematic bias of almost 8 ppb was observed for CO. A re-evaluation of the calibration cylinders used on board the $Marion Dufresne$ will be carried out during the OP4 rotation in December 2023. This bias explains the systematic offset observed for CO in Figure 7. The reactive gases instruments have worked on average 90% of the 2021-2023 period. About 30% of the data have been filtered due to local pollution, using NO spikes to confirm local pollution (NO > 1 ppb), in addition to CO spikes.

The ozone seasonal variation (Fig. 8) shows a minimum ($\sim$ 18 ppb) in summer (January) and a maximum (between 30 and 40 ppb) during the winter season. This seasonal variation and associated ozone mixing ratios are in good agreement with the few ozone measurements performed in the remote mid-latitude southern Hemisphere, Amsterdam Island (Gros et al., 1998), and Cape Grim (Parrish et al., 2016). Especially, the seasonal variation observed in 2022 (with a maximum of 30 ppb) is very similar to the mean seasonal cycle from the long time series obtained at Cape Grim since 1982 (Schultz et al., 2017), show-

ing the representativity of these measurements for this region. The higher winter values measured during 2022 correspond to campaigns performed at latitudes higher than 30°S and may have been influenced by the impact of biomass burning, which are active in Southern Africa during the May-November period. The same seasonal behavior is clearly seen for $CH_4$ and CO (Fig. 7), with minimum observed in summer (February-March) due to the photo-oxidation of this species (OH sink). The high concentrations of $CH_4$, CO and $CO_2$, observed in winter 2022 confirms the continental origins of this anomaly. It is also

important to note the strong increase of $CH_4$ between summer 2021 and summer 2022, by about 21 ppb, in line with measurements taken at Amsterdam Island and other observatories (Lin et al., 2023). The spatial variability of reactive gases is shown





on the trajectories of the $MarionDufresne$ in Figure 9. It is important to note that this representation also incorporates the seasonal variability. As expected, the highest concentrations are generally measured near continents. High concentrations were measured on the road north of Madagascar ($CH_4 \sim 1850$ ppb, $CO_2 \sim 415$ ppb, $CO \sim 70$ ppb, and $O_3 \sim 40$ ppb) or more

occasionally off the coast of South Africa ($CH_4 \sim 1900$ ppb, $CO_2 \sim 420$ ppb, $CO \sim 90$ ppb, and $O_3 \sim 45$ ppb). The low concentrations are spatially more variable and probably linked to meteorological conditions. In pristine condition, the lowest concentrations reached in 2021 were 1800 ppb, 410 ppb, 30 ppb, and 10 ppb respectively for $CH_4$, $CO_2$, CO and $O_3$. As expected, the most reactive species ($O_3$, CO) present a more pronounced spatial variability than for the greenhouse gases $CO_2$ and $CH_4$. We also note on Figure 9 high and continuous concentrations of CO ($> 50$ ppb) and ozone ($> 25$ ppb) on certain

paths far from the main anthropogenic sources. This corresponds to long-range transport; usually of biomass burning plumes. For example, on the Crozet-Kerguelen route of the SWINGS campaign, several concentration peaks were detected between February 11 and 17, 2021; i.e. an increase in $CH_4$ of around 6 to 12 ppb, CO of 5 to 10 ppb, and $CO_2$ of 10 to 20 ppm.

### 5.3 Atmospheric aerosols

#### 5.3.1 In-situ aerosols

Figure 10 shows the mean and quartiles of aerosol and CCN concentrations measured by the CCN-100, SMPS, OPC-N3, and CPC during the various campaigns in which the instruments operated correctly. The mean total aerosol number concentration varied between 490 $cm^{-3}$ (average during SWINGS on Jan.-Feb. 2023) and 1200 $cm^{-3}$ (average during MAYOBS on Sept. 2021) (Figure 10), which is consistent with the orders of magnitude reported in the literature for marine regions in clean air masses of the Southern Hemisphere (Humphries et al., 2021; Sellegri et al., 2023). In the coarse mode (OPC-N3), 50 % of the

1-10 μm particle concentrations are in the range 1 $cm^{-3}$ and 12 $cm^{-3}$.
We note a strong spatial variability of the aerosol concentration, with low concentrations on the northeastern parts of the ship's track (Fig. 11) and higher concentrations north of Madagascar and southeast of the African continent. The high concentrations observed north of Madagascar correspond to high CO, and $O_3$ concentrations also observed (see section 4.2), pointing to potential terrestrial outflows, but they also match with high concentrations of $CH_4$ and picophytoplankton cell abundances

and therefore a biological influence can not be excluded. Below latitude $40°$S, the total aerosol concentration is also spatially variable, probably linked to a greater variability of meteorological conditions (storms and strong swells) and potentially also due to a high variability of phytoplankton concentrations on the subantarctic front region (Fig. 3) that need further investigation. Figure 12 shows the particle size distribution of the aerosols throughout the measurement period, fitted by log-normal functions (size ranges 20 nm - 350 nm and 400 nm - 6 μm) in three modes classically observed in the atmosphere, corresponding to

the Aitken, accumulation, and coarse modes. The calculated median diameters are at 29.7 nm, 108.7 nm, and 1.67 μm and the standard deviations are 1.63, 1.67, 1.54 for the Aitken, accumulation, and coarse modes, respectively (Table 2). A 2nd coarse mode can also be observed at 4.5 μm but is at the limit of validity of OPC-N3. Similarly, there may be a second undetected accumulation mode between 300 nm and 400 nm due to instrumental limitations. The size distributions were then separated based on the wind speed measured on the ship. At a wind speed less than 10 $m\,s^{-1}$, the primary marine aerosol emission is



**Figure 7.** Time series of $CO_2$, $CH_4$, and CO concentrations monitored on board the $Marion Dufresne$ (blue circles) compared to Amsterdam Island observatory (red line). The gray points correspond to measurements on board $Marion Dufresne$ during stopovers in a port, or contaminated by the ship exhausts. Each point represents an hourly mean. The CO peak in February 2021 corresponds to a fire on Amsterdam Island.

considered low, in the range 10-20 m s$^{-1}$ the primary emission becomes significant, and for storms ($> 20$ m s$^{-1}$) very few observations have been made on ships (e.g., Ovadnevaite et al., 2014; Bruch et al., 2021).



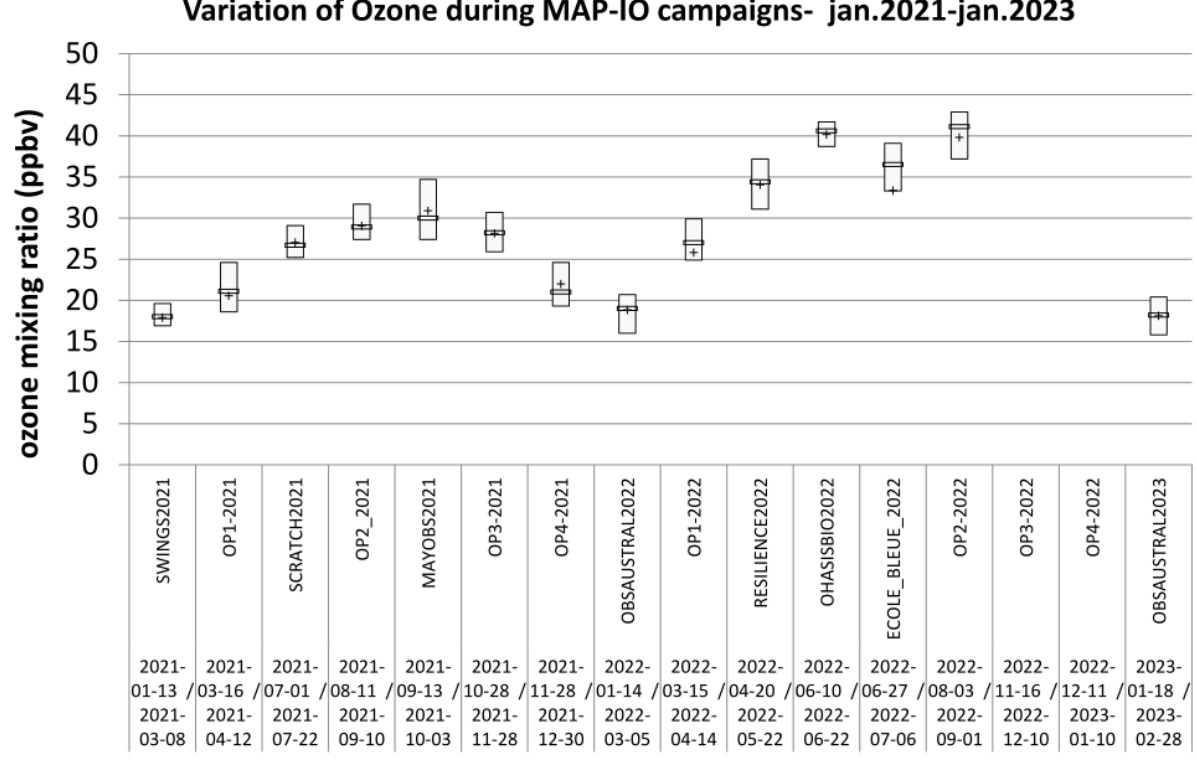

**Figure 8.** Mean and quartile concentration of ozone (ppb) during the MAP-IO campaigns - January 2021 to January 2023.

The aerosol concentration in the Aitken mode is significantly higher under low wind conditions ($< 10 \mathrm{~m~s^{-1}}$). The result is reversed for the accumulation and coarse modes where the highest concentrations are measured for strong winds. For the coarse mode, concentrations are 4 times higher for strong winds ($> 20 \mathrm{~m~s^{-1}}$) than for low winds ($< 10 \mathrm{~m~s^{-1}}$) which is in line with the production of primary marine aerosols by wave breaking. We also note in figure 12 that 50% of Aitken mode concentrations are below the mean value for low and moderate wind speeds. This result shows the great variability of concentrations in this mode, with strong concentration peaks observed in a quarter of the valid data.

With the exception of the Aitken mode, the median diameters are very close for low and moderate wind speeds, while the median diameter is systematically lower for high wind speeds, which may indicate an impact of condensation growth that is higher for low and moderate winds than for high winds. Average diameters also vary considerably with wind speed. Therefore it will be relevant to look for other factors than wind speed (SST, precursor gases, aerosol wash out along the back-trajectory path, etc.) which significantly influence emissions, formation and growth of submicron aerosols.

The mean CCN concentration at 0.2 % of sursaturation ($CCN_{0.2}$) ranged between 120 $\mathrm{cm^{-3}}$ (SCRATCH, July 2021) and 300 $\mathrm{cm^{-3}}$ (MAYOBS, Sept. 2021). The variability is particularly important on CCN concentrations with 25% of the concentrations being lower than 70 $\mathrm{cm^{-3}}$ in July 2021 and higher than 450 $\mathrm{cm^{-3}}$ during Sept. 2021. We also note that for campaigns with



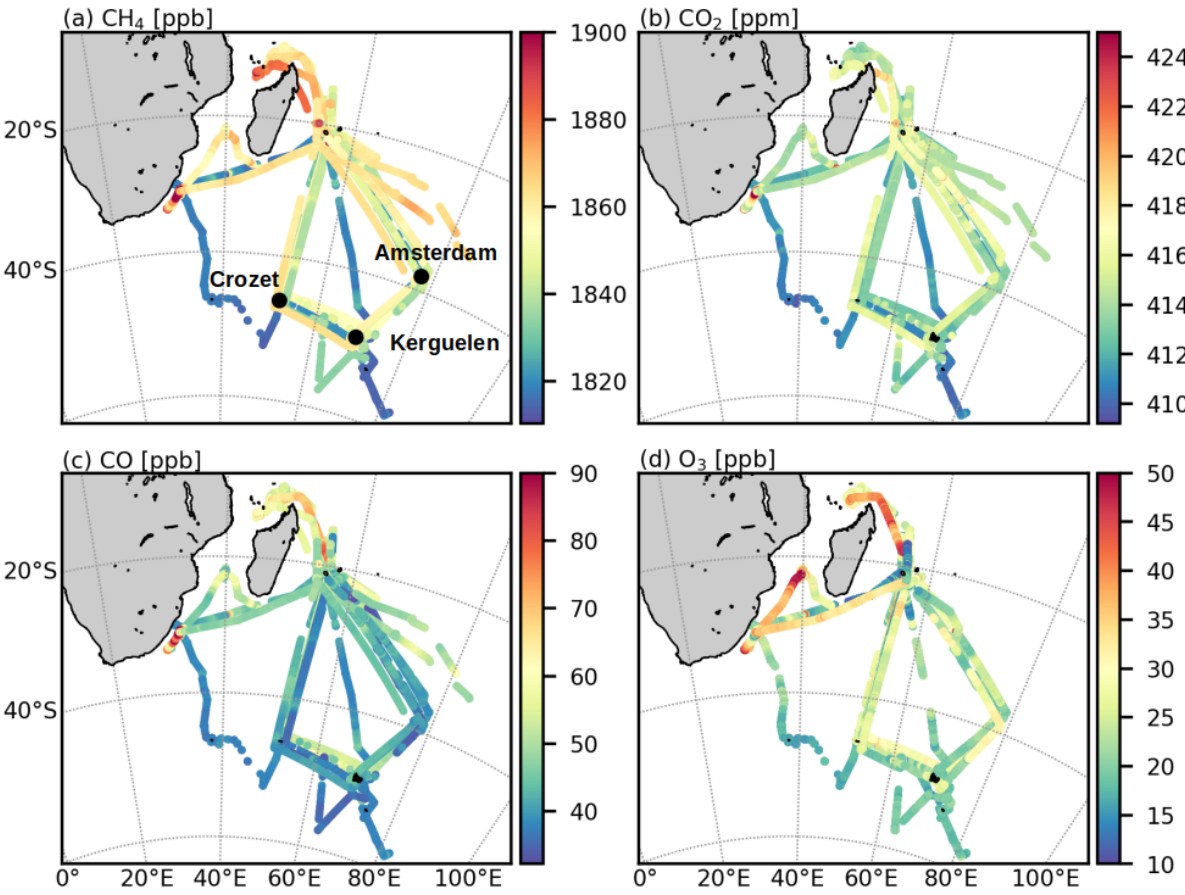

**Figure 9.** Variation of gaseous species mixing ratios during the 2021-2023 MAP-IO campaigns. The black dots represent the position of Crozet, Kerguelen, and Amsterdam Island.

similar routes, the average $CCN_{0.2}$/CN ratios, which are a function of aerosol size and composition, can be very different from one campaign to the other, while being measured during the same season. For example, this ratio was 0.18 for Jan-Feb 2021 and 0.5 for Jan-Feb 2023 indicating that during in Jan-Feb 2021, the $Marion Dufresne$ may have passed through air masses composed of more hydrophobic and/or smaller aerosols.

### 5.3.2  Aerosol (Remote Sensing)

From early July 2021 to early May 2022, $\sim$ 10000 cloud-free AOD (level 1.5) have been recorded. In cloud-free conditions, AOD are measured every 15 min and 3 min during daytime (70% of records) and nighttime (30 % of records), respectively. The mean AOD is 0.095 $\pm$ 0.070, 0.086 $\pm$ 0.05, and 0.060 $\pm$0.03 at 440, 500, and 870 nm, respectively. The mean Angström Exponent (AE), computed between 440 and 870 nm, is 0.7 $\pm$ 0.3 and the mean water vapor content 2.8 $\pm$ 1.1 g cm$^{-1}$. The





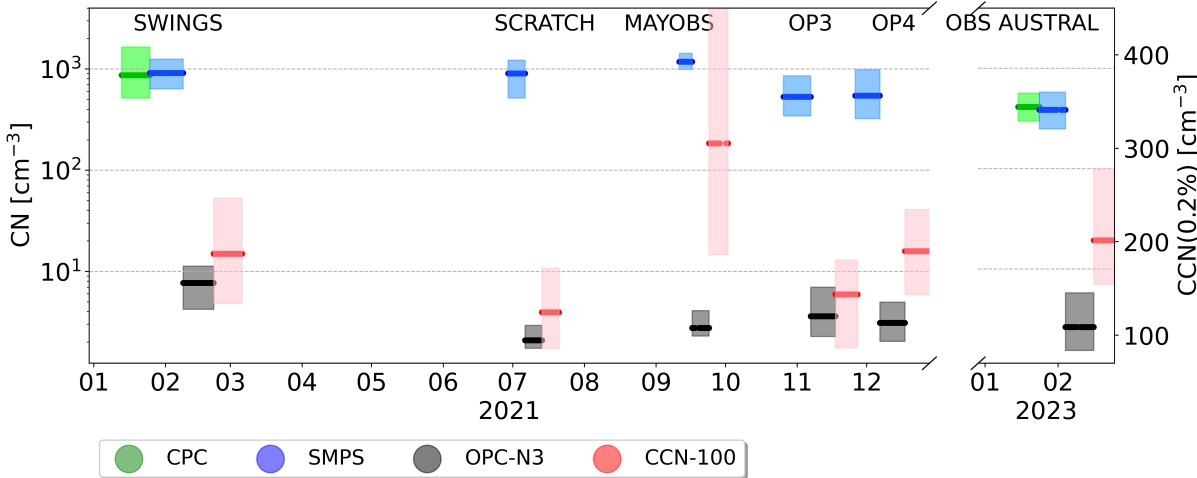

**Figure 10.** Mean and quartile concentration of aerosols (in $\mathrm{cm}^{-3}$, scale at the left) measured by the SMPS (blue), CPC (green) and OPC-N3 (black). The CCN at 0.2% of supersaturation measured by the CCN-100 was superimposed (in $\mathrm{cm}^{-3}$, scale at the right).

**Table 2.** Aerosol size distribution of each mode fitted into a lognormal distribution (total number (N), mean diameter (D), standard deviation ($\sigma$) and root mean square deviation (RMSE).

| Wind speed | $< 10 \, \mathrm{m \, s^{-1}}$ | $10\text{-}20 \, \mathrm{m \, s^{-1}}$ | $> 20 \, \mathrm{m \, s^{-1}}$ | Average |
|---|---|---|---|---|
| Aitken mode | D = 28.0 nm | D = 32.0 nm | D = 26.5 nm | D = 29.7 nm |
| | $\sigma$ = 1.71 | $\sigma$ = 1.49 | $\sigma$ = 1.60 | $\sigma$ = 1.63 |
| | N= 886 $\mathrm{cm}^{-3}$ | N = 500 $\mathrm{cm}^{-3}$ | N = 600 $\mathrm{cm}^{-3}$ | N = 723 $\mathrm{cm}^{-3}$ |
| | RMSE = 65.1 $\mathrm{cm}^{-3}$ | RMSE = 38.4 $\mathrm{cm}^{-3}$ | RMSE = 10.6 $\mathrm{cm}^{-3}$ | RMSE = 56.2 $\mathrm{cm}^{-3}$ |
| Accumuation mode | D = 109.5 nm | D = 109.0 nm | D = 82.0 nm | D = 108.7 nm |
| | $\sigma$ = 1.69 | $\sigma$ = 1.60 | $\sigma$ = 1.71 | $\sigma$ = 1.67 |
| | N = 216 $\mathrm{cm}^{-3}$ | N = 170 $\mathrm{cm}^{-3}$ | N = 326 $\mathrm{cm}^{-3}$ | N = 205 $\mathrm{cm}^{-3}$ |
| | RMSE = 6.85 $\mathrm{cm}^{-3}$ | RMSE = 4.27 $\mathrm{cm}^{-3}$ | RMSE = 3.6 $\mathrm{cm}^{-3}$ | RMSE = 6.0 $\mathrm{cm}^{-3}$ |
| Coarse mode | D = 1.66 $\mu$m | D = 1.70 $\mu$m | D = 1.56 $\mu$m | D = 1.67 $\mu$m |
| | $\sigma$ = 1.59 | $\sigma$ = 1.52 | $\sigma$ = 1.45 | $\sigma$ = 1.54 |
| | N = 2.35 $\mathrm{cm}^{-3}$ | N = 5.02 $\mathrm{cm}^{-3}$ | N = 8.72 $\mathrm{cm}^{-3}$ | N = 3.65 $\mathrm{cm}^{-3}$ |
| | RMSE = 0.86 $\mathrm{cm}^{-3}$ | RMSE = 0.87 $\mathrm{cm}^{-3}$ | RMSE = 0.96 $\mathrm{cm}^{-3}$ | RMSE = 0.89 $\mathrm{cm}^{-3}$ |

average AOD is very consistent with Mallet et al. (2018). Overall, the values of AE range from about 0.0 to $\sim$ 1.5 with a mean of $\sim$ 0.7. Such values are typical of marine aerosols and consistent with Mallet et al. (2018) and also with Smirnov et al. (2002) who report AE ranging from 0.3 to 0.7 over clean marine regions free of continental influences. In the following we address the

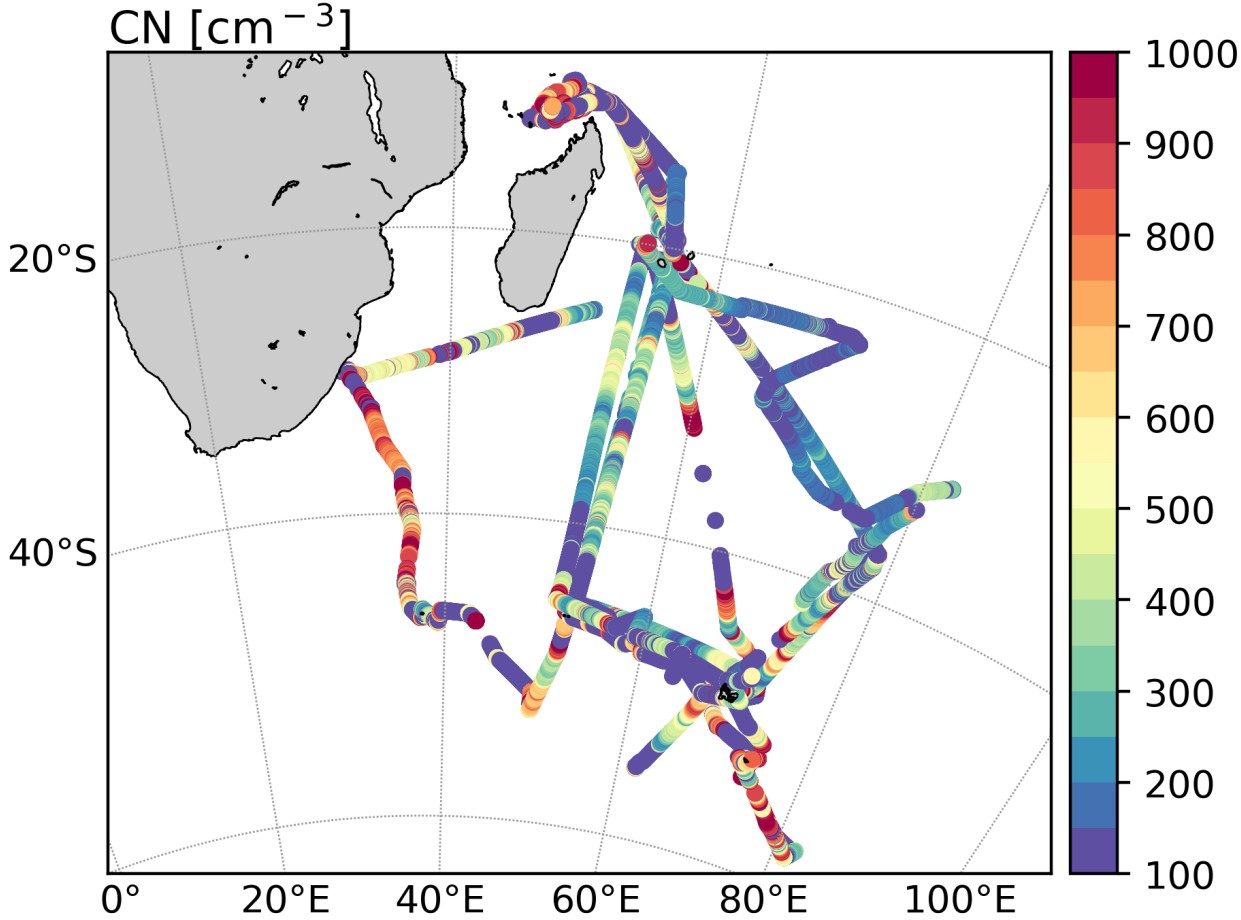

**Figure 11.** Evolution of the total number of aerosols (CN in $\mathrm{cm}^{-3}$) along the path of the $Marion Dufresne$ over the year 2021 and between January and March 2023. Note that the SMPS has been under maintenance in 2022. The discontinuous zones correspond to the filter on the relative wind (from rear direction or at low speed) or during stops of the ship in order to eliminate any risk of contamination by the ship's exhaust or by the activity on board. To limit the superposition of the trajectories, they were slightly offset.

spatial variability, in terms of latitudinal variation, for the AOD, AE, and water vapor. For that purpose, we average all the data recorded within $1°$ latitude grid (Fig. 13). For the sake of clarity, we organized the results by season, respectively labeled DJF (December to February), MAM (March to May), JJA (June to August), and SON (September to November). Clear decreasing trends are observed from the North to the remote South part of the Indian Ocean, for AOD and water vapor. This is observed for all seasons. The reported AOD values are consistent with the results from Mallet et al. (2018) derived in the South Indian Ocean. An interesting behavior is the separation of relatively low AOD (lower than 0.075) and relatively high AOD ($> 0.08$) which occurs around $35°$ in latitude in DJF, $25°$ in MAM, $20°$ in JJA, and $30°$ in SON. This latitudinal behavior suggests that

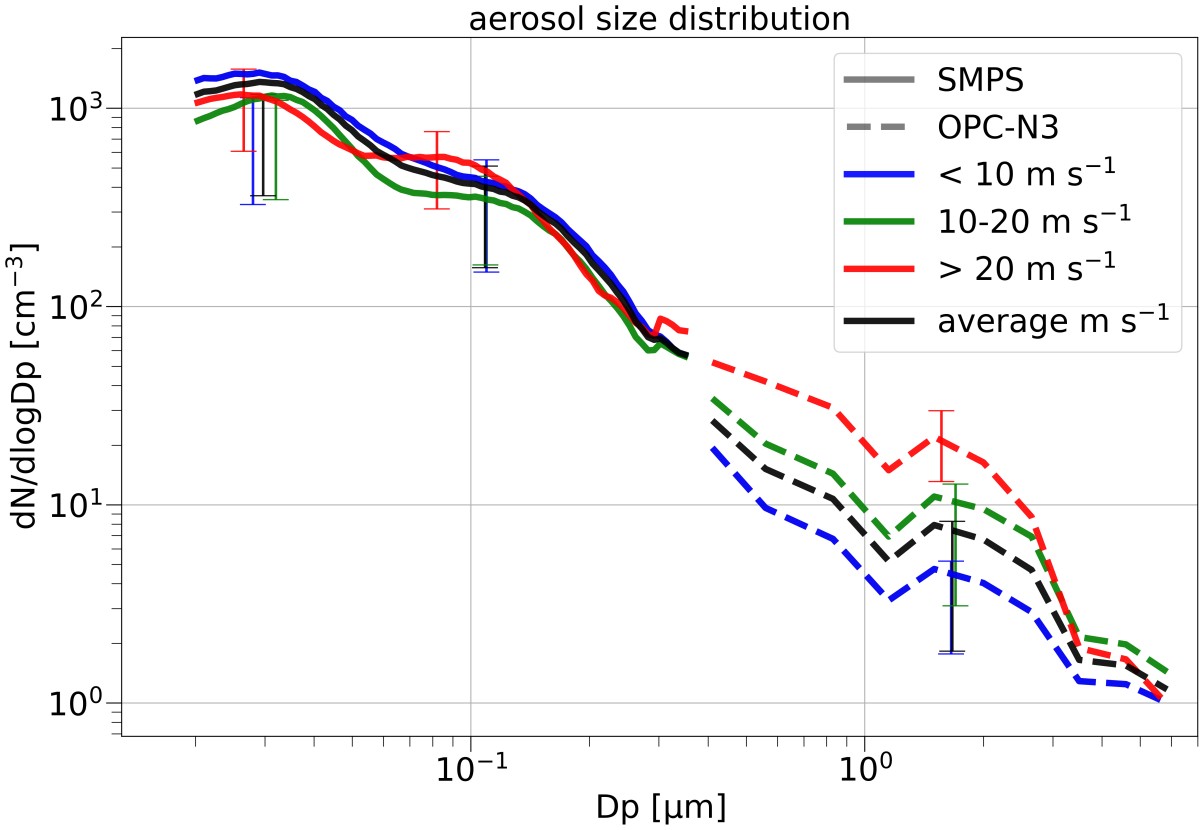

**Figure 12.** aerosols size distribution and root mean square deviation observed during 2021 and between January and March 2023 observed by the SPMS and OPC-N3. Vertical lines corresponds to the quartile concentration measured at the bin corresponding to each mean diameter of the log-normal functions.

this separation follows the motion of the ITCZ. According to Mallet et al (2018), the relatively low AOD values reflect the presence of sea salt while the higher values are more typical of sulfate. These values are also consistent with those reported by Mascaut et al. (2022) around Reunion Island. The same comment can be made for the Angstrom Exponents, whose values are typical of marine aerosols. The increase in absolute humidity with the latitude is not surprising as one travels closer and closer to the Equator. These first results are also consistent with those obtained during the AEROMARINE field campaign, around

Reunion Island (Mascaut et al., 2022).

### 5.4 Comparaison IWV (GNSS / SAOZ / Photometer)

Water vapor, a key climatic constituent, is a challenging variable to measure due to its spatial and temporal variability (Bock et al., 2013).The integrated vertical column of water vapor can be assessed by different instruments co-located onboard the $Marion Dufresne$. The GNSS instrument performs measurements over a large atmospheric cone above an elevation at the



**Figure 13.** Latitudinal variation of AOD (a), Angström Exponent (b), and Water vapor content (c) for each season (DJF, MAM, JJA, SON). The observations considered here are coming both from day and nighttime records. "Error bars" stand for the standard deviation of the corresponding properties within the 1∘ latitude grid.





horizon of 3 degrees over the whole sky covered by GPS satellites (360°) (Bosser et al., 2022). Other instruments are a UVVis Mini-SAOZ spectrometer and a Sun photometer. The predecessor of Mini-SAOZ has shown consistency with GNSS observations during the DEMEVAP campaign in September-October 2011 at the northern hemisphere mid-latitude station of OHP (Observatoire de Haute Provence) in France (Bock et al., 2013). Mini-SAOZ measures a slant column changing as a function of the Sun's position. The spatial extent of air masses sampled by the Mini-SAOZ could be associated to a 2-D

polygon projected to the surface (Garane et al., 2019). Vertical columns are obtained then divided by the corresponding AMF for $H_2O$. Only observations at SZA lower than 60°, sensitive to tropospheric constituents, are used in this study. Measurements presenting a computed color index (ratio of the fluxes at 550 and 350 nm already corrected from constituents) higher than 5 were filtered. In the case of the photometer, it performs direct sun/moon measurements. The air masses sampled by the instrument correspond to a column in the direct direction between the sun or the moon and the instrument. Figure 14 (top panel)

shows the time evolution of IWV observed by the GNSS (black points), the Mini-SAOZ (red points) and the Sun photometer (blue points) during the $Marion Dufresne$ trips. Despite the three instruments' different sampling geometry corresponding to somewhat different atmospheric $H_2O$ amounts, their observations clearly show a strong agreement. The evolution of IWV is also correlated with the latitude of the ship (bottom panel of Fig. 14). Large amounts of IWV are observed when the ship travels in tropical regions at latitude equatorward 25°S with a mean value of $8.5 \pm 2.1$ ($1\sigma$) molec cm$^{-2}$, while mean values two to

three times weaker are observed at latitudes poleward 45°S over the Indian Ocean. Figure 15 presents the zonal mean IWV value and $\pm\sigma$ observed by the three instruments between 60°S and 10°S. The latitudinal gradient of IWV is well highlighted by each instrument. A good agreement is found within $1\sigma$ between the measurements with a higher latitudinal amplitude for SAOZ IWV.

## 5.5 UV and stratospheric ozone

The UV erythemal index (UV-E) measured by the SUV-E radiometer under clear sky conditions is shown in Figure 16. These conditions were identified based on cloud fraction measurements captured by the sky camera and a previously optimized cloud fraction threshold for other sites in the southwestern Indian Ocean region (Lamy et al., 2021b). A color code is utilized to signify the variable solar zenith angle. The total ozone column, derived from the Mini-SAOZ instrument, and the latitudinal variation are portrayed in red and blue, respectively. As the vessel moves poleward from Reunion to Kerguelen, the latitude

changes from approximately 20°S to about 48°S, the daily minimum solar zenith angle values reached is higher at higher latitudes, visually represented in Figure 16 by a gradient line color that shifts towards yellow as the solar zenith angle values increase. Conversely, the daily minimum solar zenith angle values decrease when the vessel travels toward the equator, or in other words, the sun's highest point in the sky goes to the zenith near 20° SZA as the vessel travels toward the equator. There's a notable decrease in UV observed in conjunction with the movement towards higher latitudes (from Reunion to Kerguelen).

This is anticipated to be inversely correlated with the increase in the total ozone column, generally noted in mid and high latitudes due to the BDC.
The measurements of UV-A, UV-B, and UV-E radiation taken by the radiometers SUV-A, SUV-B, and SUV-E on August 30 and 31, 2021, are illustrated as blue, green, and red dots in Figure 17, respectively. The cloud fractions, measured by the

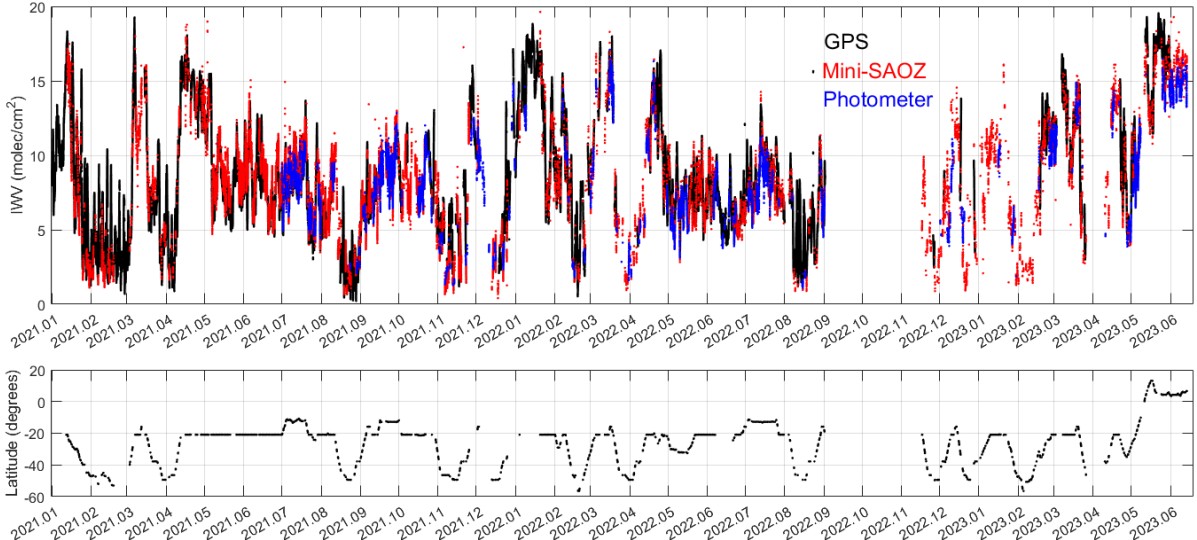

**Figure 14.** Evolution of integrated water vapor observed by GNSS (GPS), Mini-SAOZ and photometer since January 2021 (top panel). Latitude of the $Marion Dufresne$ (bottom panel). The interruption of measurements at the end of 2022 corresponds to the technical stop of the vessel in Singapore.

camera, are also denoted by black points. The mean daily values of total ozone were similar for both days, 311.81 Dobson Units
(DU) on August 30 and 317.20 DU on August 31. However, the cloud fraction diurnal values presented significant disparities. Given the proximity in total ozone values, one might expect similar measurements of UV radiation for both days. However, this was not the case due to the large variations in cloud conditions. On August 30, the day was predominantly overcast, with high cloud fraction (CF) values persisting between 03:00 and 09:00 UTC. Conversely, August 31 was a partially clear day, characterized by intermittent low to moderate CF values (between 0.1 and 0.5) around 3, 4, 6, and 8 am. During phases of very
low cloudiness, UV radiation increases until solar noon time (lowest solar zenith angle of the day), and then decreases, thus producing a bell-shaped curve. An increase in CF values to approximately 0.5 on August 31, particularly around 6 and 8 AM (close to local solar noon), were anti-correlated with a decrease in UV-A, UV-B, and UV-E values. High CF values on August 30 were associated with substantial attenuation of UV-A, UV-B, and UV-E radiation throughout the day. UV-B is less affected by clouds than UV-A due to the spectral dependence of clouds transmittance. Past research has shown that cloud transmittance
is higher for UV-B than for UV-A, approximately 60% and 40% respectively (Seckmeyer, 1996).

## 6   Conclusions

The purchase and deployment of the instrumentation on the vessel took place between September 2020 and January 2021. Despite all the often complex operational and technical difficulties to be resolved on an oceanographic vessel (i.e. mechanical,



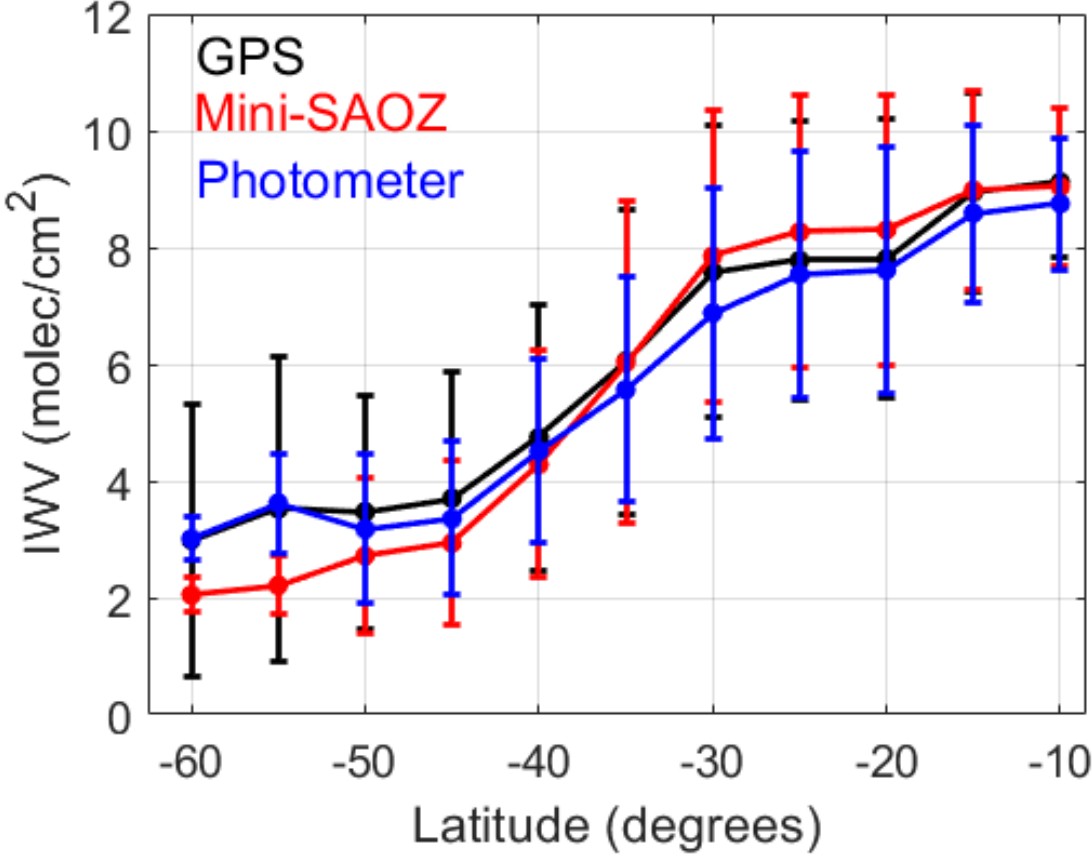

**Figure 15.** Zonal IWV mean $\pm 1\sigma$ observed by GNSS (GPS), Mini-SAOZ, and Photometer instruments on board the $Marion Dufresne$. GNSS and Photometer data were sampled at Mini-SAOZ time.

instrumentation, IT, work on the ship's hull), and the difficult context of the COVID pandemia (PPE equipment, limited contact
with sailors on board), nearly 700 days of observation on sea were carried out between January 2021 and June 2023.
The first climatological studies based on the database show significant intra-seasonal and latitudinal variabilities of greenhouse
gases concentration and aerosol optical depth. The total number, the size distribution, and the CCN properties of aerosols vary
considerably upon the air masses sampled (from a hundred to thousand particles per $cm^{-3}$). It can be seen that the average
CCN$_{0.2}$/CN ratios varies significantly for a given geographic location and season, reflecting important variations in the hygro-
scopic and/or size distribution of the aerosols measured. The first rough level of investigation of the factors influencing the
aerosol concentration and size show that wind speed explains a significant part of the coarse mode aerosol concentrations, but
other factors are needed to explain the submicron aerosol concentrations.
The phytoplankton community structure has been studied by resolving several size classes and the collected data sets can be
considered as one of the most important dataset of surface phytoplankton abundance collected so far in such a short time and

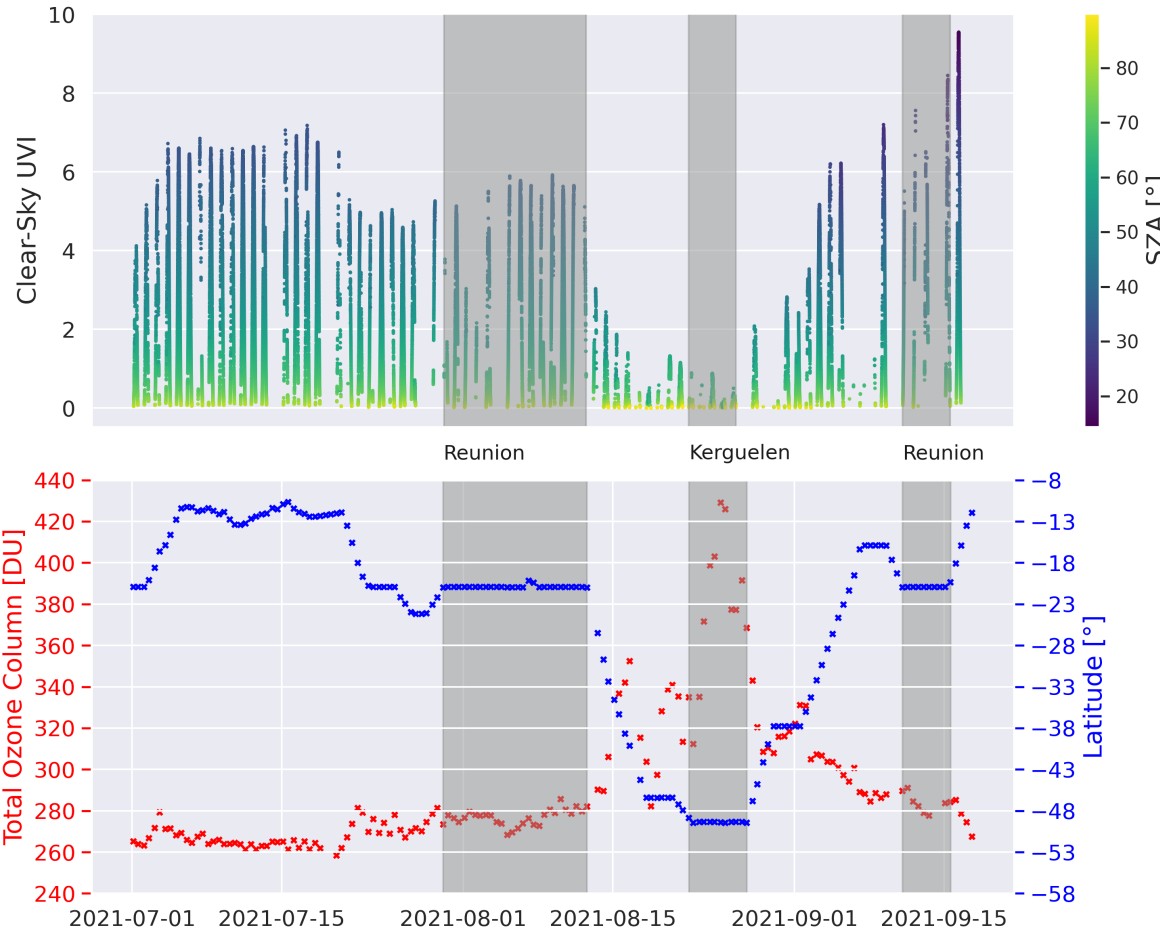

**Figure 16.** Daily Clear-sky UV-E from SUV-E radiometer with color code indicating the SZA (top panel) and twilight total ozone column and latitude from Mini-SAOZ instrument (bottom panel) between 01-07-2021 and 15-09-2021. The gray zones indicated the period where the $Marion Dufresne$ ship was at the Reunion and Kerguelen stations.

in this remote area. The single cell study evidenced physiological and possible taxonomic differences following latitudes more than seasons, by the shift in size per functional phytoplankton group resolved. A deeper study into the contribution of each group to chlorophyll A and carbon will enhance the understanding of the role of phytoplankton in sustaining biogeochemical cycles and the trophic web. Further insight into this representative ocean/atmosphere data set will open the path to some strong statistical relationships that would raise new scientific questions.

The collection of the in-situ data of MAP-IO under different latitudes and seasons, sea state, and meteorological conditions should provide an original framework for studying and improving the parameterizations of turbulent fluxes at the surface. For example, machine learning methods exploiting large databases could be used. In this context, the vast instrumental setup


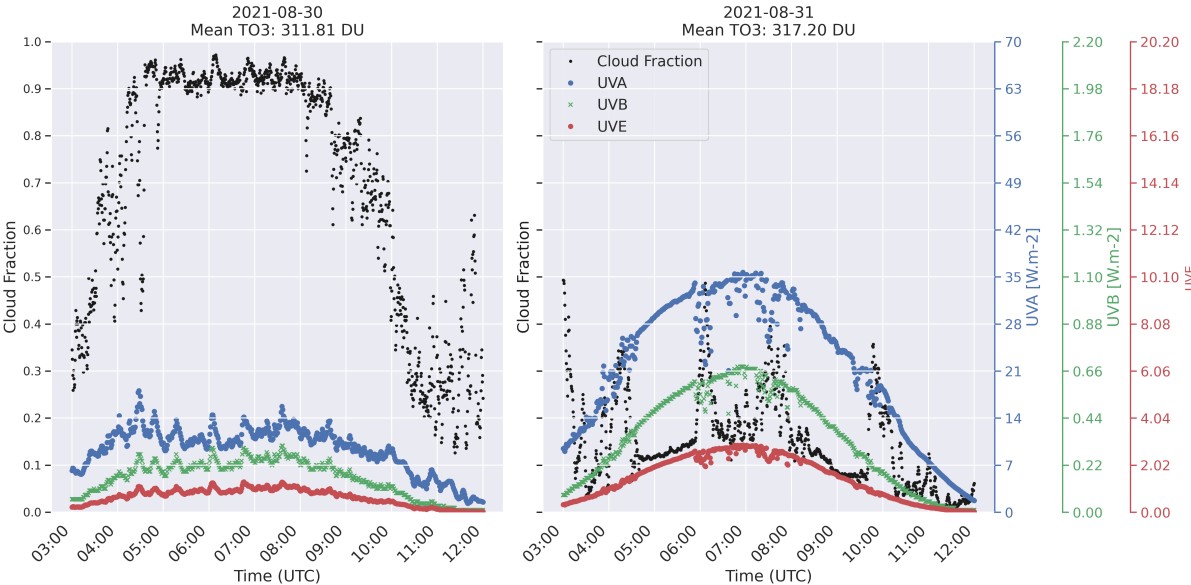

**Figure 17.** UV-A, UV-B, UV-E and cloud fraction measurements taken on the days of August 30 and 31, 2021.

installed onboard to characterize the properties of aerosols, gas traces, and phytoplankton class should allow us to evaluate and improve parameterization of ocean-atmosphere exchanges, especially under strong winds and high swell conditions. Inte-

grated optical depth measurements of ozone, UV, and aerosols over the Indian and Southern Ocean can be integrated for the calibration-validation of the European mission EARTHCARE (including the ATLID lidar and MSI spectrometer), the USA AOS mission (Atmosphere Observing System, including polarimeter, lidar, and water vapor sounder), the European EPS-SG missions (Eumetsat Polar System - Second Generation, which will include the 3MI polarimeter, IASI-NG and METIMAGE radiometer), and SENTINEL 2, 3, and 5P by sampling different atmospheric conditions. Measurements by class of phytoplank-

ton will make it possible to contribute to a better Plankton functional groups community structure quantification by dedicated remote sensing ocean color products (Uitz et al., 2006; Alvain et al., 2008; El Hourany et al., 2019) (e.g. OLCIA, OCLIB, GlobColour) over the Indian and Southern Oceans.

Several innovations have been developed. The development of fast GNSS zenith delay inversion algorithms for integrated water vapor content restitution for integrating the specificities of ship movements. Adaptations to make the CE318T lunar/solar

photometer autonomous and adapted to heavy swells and icing were also developed. The Mini-SAOZ, successfully tested for the first time on a ship, also enabled continuous acquisition of integrated water vapor and ozone columns over the ocean. These three instruments provide reliable, continuous, and local measurements of atmospheric water vapor in regions where the main source of observations is space-based. This measurement, performed routinely with low latency, could contribute to the evaluation of, or even assimilation into, numerical weather models. Long-term monitoring of the spatial and temporal distribution

of water vapor could contribute to the monitoring of climate change in these regions.



Test campaigns for onboard instruments, such as the CE370 micro-lidar during the AMARYLLIS-AMAGAS campaign, demonstrated the potential for the permanent deployment of active remote sensing instruments on ships. Generally speaking, the majority of instruments have operated autonomously or with limited human intervention. Many of the data can be transferred within 10 minutes and thus open up the prospect of near-real-time operation. These successful proof-of-concepts
open up interesting prospects for the development of operational observations both for international research networks such as AERONET or NDACC and for operational numerical forecasting networks. If pursued over future years, the MAP-IO program could be a useful tool for monitoring climate trends, for calibrating and validating numerical forecasting models, and space sensors in a region almost devoid of observation.

## 7  Data availability

Atmospheric data are available on the AERIS datacenter: https://www.aeris-data.fr/ (last access: 8 November 2023). Cytometry data are available on the SEANOE datacenter: https://doi.org/10.17882/89505 (Thyssen et al., 2022b)

**Appendix A:  MAP-IO instrumentation on-board**

*Author contributions.*  PT is the head of the MAP-IO program. NM, JMM, LG, FRL have been in charge of the installation and the maintenance of the instruments on-board. DM, OP, GP, GD, GA, and EN performed and maintained the IT system of the program. PT, KS, MD,
JB are responsible for the aerosols in-situ data and have contributed to the analysis and the figures of the paper. SP is the program's administrative manager. PB is in charge of the GNSS component and is responsible for the routine analysis of the GNSS data acquired from the research vessel. OM, OQ, SH, LG organized the operation of the $Marion Dufresne$ and helped us with the installation and maintenance of the instruments on the ship. LB, GD, PG, and BT were in charge of the part aerosol remote sensing. LB and GD helped with the installation of ship born photometers and assured the quality control of the measurements. PG and BT contributed to the analysis of tendencies and figures
of the paper. MS and VD gave support to the installation of the remote sensing instrumentation on the ship and participated in the definition of the strategy to transform the program into an observatory. MR, MD, and ML are responsible for the GHG in-situ measurements: data treatment, logistic, and scientific analysis. LG, NM, JK, and MT supervised the installation of the Cytosense onboard the $Marion Dufresne$. MT and JK are responsible for the Cytosense scientific operations and instrument maintenance. LG was the technical responsible of the Cytosense deployment, quality control, and data acquisition. MT analyzed and interpreted the data set. KL analyzed the collected images
from the Cytosense. JP worked on the analysis of the aerosol size distributions and on the homogenisation of the paper's figures. AP and MNP were in charge of the measurements of total $O_3$, $NO_2$, and $H_2O$ columns by the Mini-SAOZ spectrometer. MNP supervised the installation of Mini-SAOZ onboard the $Marion Dufresne$ and assured the quality control of the measurements. KLa was in charge of the UV calibration, quality control, and analysis. AP contributed to the analysis of ozone and UV variability and $H_2O$ intercomparisons. PB, OB, and JVB were in charge of the GNSS system deployment. PB performed the corresponding analysis and observation comparisons. AC and
VG were in charge of the measurement of in-situ ozone and $NO_x$: data treatment and scientific analysis. All authors have read and agreed to the published version of the manuscript.

**Figure A1.** Images of the different instruments on-board. Top: photometer, GNSS antenna, Mini-SAOZ and the meteorological station. Middle: flow cytometer, all sky camera / UV radiometers and calibration gas bottles. Bottom: gases and aerosols analysers on the vibration-dampened table and inlets on deck i. Photographies credits T. Portafaix, F. Rigaud-Louise and J.M. Metzger.



*Competing interests.* The contact author has declared that none of the authors has any competing interests.

*Acknowledgements.* Authors highly acknowledge the TAAF, IFREMER, LDAS and GENAVIR for their help and constant support in the installation and the maintenance of all scientific instruments on board the $Marion Dufresne$. The authors also thank the technical teams of the LACy and OSU-R engaged in the data acquisition and the maintenance of the instruments of the MAP-IO program and the financial and human support of each laboratories partners such as OSU-R, LACy, LaMP, LAERO, LOA, LATMOS, LSCE, MIO, and ENTROPIE. MAP-IO is a scientific program led by the University of La Réunion (LACy and OSU-R) and was funded by the European Union through the ERDF programme, the University of La Réunion, the SGAR-Réunion, the Région Réunion, the CNRS, IFREMER, and the Flotte Océanographique Française. The authors from LACy acknowledge the support of the European Commission through the REALISTIC project (GA 101086690). Olivier Bousquet helped us to install the GNSS antenna. A special thank you to Gérald Gregory who helped us build the oceanographic component of this program.



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
