# Peer review of "MAP-IO, an atmospheric and marine observatory program onboard MarionDufresne over the Southern Ocean"

_Earth System Science Data, 2023_

## Author Comment (AC2)

**Example of SMPS filtered by the PI**

This figure shows an example of data filtered by the PI. On the left, the SMPS size distribution on the 19 january, 2021. Strong peaks of concentration still present after the dynamical and chemical filters. These concentration are homogeneous upon a large range of bins. On the right the data used in the paper.

By experience, this sudden increase does not have the shape of a nucleation peak and can only come from a local source of pollution from activities on the ship. The PI therefore added a warning flag to the dataset and these data were not used in the article.

[Figure]

**OPC-N3 comparisons**

The monthly average of PM2.5 mass concentrations of the 3 OPC-N3 were compared over one year (June 2021 to June 2022).

Apart from OPC-N3 N°3 in October 2021 which underestimates the PM2.5 concentration by 20 µg/m3 compared to the other two instruments, the difference between the three instruments is less than 10 µg/m3. The average relative errors are 1.7% between OPC-N3 N°1 and OPC-N3 N°2, 9% between OPC-N3 N°1 and OPC-N3 N°3 and 7.3% between OPC-N3 N° 2 and OPC-N3 N°3.

In the paper we have used the data of the OPN3 N°1 which is close to the data of the OPC-N3 N°2 (see below).

These explanations and the figure have been added as supplement.

[Figure]

---

## Author Response (AR2)

**Referee 1:**

*Tulet et al present an observational dataset from a shipborne campaign (MAP-IO) over the Southern Ocean from Jan 2021-June 2023. I want to congratulate the team for overcoming many challenges to make the campaign possible. This is a valuable dataset because it contains many important parameters (such as climatology, gases, aerosols, and phytoplankton) over the Southern Ocean where lacks of observations. The authors present some interesting results and potential research that can be explored with the data. However, I would expect more information on data quality and data processing because this is a data description paper, and that is why I recommend a major revision. Besides that, I think the authors should improve the language and text clarity, for which I have listed some suggestions below.*

We would like to thank the reviewer for its work, which we believe has significantly improved the article. We have done our best to answer its questions and recommendations.

We would like to draw your attention to an error in the post-processing of the SMPS and CPC data which has impacted the results presented in figures 10, 11, 12 and table 2. The new results do not contradict the previous version of the article, but are sufficiently different to partially modify the section 5.3.2. The instrument's post-processing algorithm has been checked several times and we are now confident in the reliability of the results. We sincerely apologize for this computation error.

*Major comments:*

> *Data quality: The authors should provide more information (particularly for the instruments in section 3.3) on instrument calibration, precision, measurement frequency, and how temperature, humidity and air pressure impact the measurement. Some information can be provided in the supplementary materials.*

We agree that some information was missed particularly for in-situ measurements. We added complements within the text, as follow:

Inlets and room acquisition:

After L174: "The inlets of the aerosols and gases instruments are located on deck "i". Aerosol analysers are installed downstream of a dedicated inlet equipped with a Nafion dryer (RH < 4 %) , and temperature and water vapour sensors.

A dispatcher distributes the sampled flow to instruments.  Inlets are designed to have a constant sampling flow rate, and thus there is no variation of pressure that could influence the measurement.

The instruments are operated in an air-conditioned room where the pressure, the humidity, and the temperature are controlled and recorded. The aerosols analyzers are equipped with an inlet line filter made of Teflon with a pore size of 5 um. The filters are changed quarterly."

Greenhouse gases:

After L193:

"Before deployment on the Marion-Dufresne ship, the Cavity Ring-Down Spectrospcopy analyzer was characterized through a battery of standardized tests designed by the ICOS Atmospheric Thematic Center (Yver-Kwok et al., 2015). These tests were performed in August-September 2020 and showed no significant dependence of $CO_2$, $CH_4$, CO concentrations on either atmospheric temperature or atmospheric pressure for the MAP-IO instrument. As far as water vapor is concerned, it is essential to correct it very precisely to obtain accurate dry-mole fraction measurements. A Nafion dryer is used to reduce the influence of water vapor and a correction is proposed by the analyzer manufacturer, applicable to all analyzers (Rella et al., 2013). This correction makes it possible to achieve WMO accuracy targets (+/-0.1 ppm for $CO_2$, and +/-2 ppb for $CH_4$) for water vapor concentrations of up to 1%."

After L197: "The calibration scale has a concentration range from 396 to 472 ppm for $CO_2$, from 1760 to 1960 ppb for $CH_4$, and from 25 to 374 ppb for CO."

O3 and NOx:

After L208: "In the event of condensation, the residual liquid water is collected in a flask connected to this manifold. As recommended in the standard operating procedures, PFA-Teflon tubing is used, as it has  a smooth (not prone to adsorption), non-porous (low absorption & diffusion), and inert (low reactions) surface.

For aerosols:

After L219:  "The complete scan on the various SMPS bins takes 5 min. The SMPS instrument can regularly be inter-compared with the SMPS from the Maido facility, which is itself regularly inter-compared within ACTRIS infrastructure (https://www.actris-ecac.eu/). "

After L228: "The choice made for the MAP-IO program was to retain supersaturations at 0.1 and 0.2 (15 min acquisition time) and 0.3, 0.4, 0.6 and 1 (5 min acquisition time). The CCN-100 is annually calibrated upon the method described by Roberts and Nenes (2005) and Rose et al. (2008). As for SMPS, the CCN-100 is regularly inter-compared with a similar instrument located at the Maido facility."

*Data processing: The authors mentioned in several places that some data are filtered out manually by the PIs of the instruments. This doesn't sound like a good practice. The authors should provide more details on how PIs manually excluded data (criteria for each instrument).*

A final quality control (QC) based on the PIs expertise is common for the restitution of data at level 1 or 2. The PIs know the limits of their instruments and are responsible for checking the quality of the data before their distribution in the data centers. Note that no data was deleted subsequent to this last QC step. A QC flag was added to the data deemed erroneous by the PI following the recommendations of IR ICOS (greenhouse gases) or IR ACTRIS (aerosols, O3, NOx).  QC flags index are referenced in the metadata or in the file headers.

To be more precise we can add tha            t dynamical and chemical filters (NOx, CO) are sometimes insufficient to deal with potential local sources of pollution. For example, air conditioning outlets are close to the aerosols inlets. Some of the ship's activities may emit VOCs that can condense on aerosols or primary new particles (e.g. painting).  We have therefore chosen to flag any data that seem to indicate a sudden spike in concentration that we believe corresponds to local contamination.

The figure below shows an example of a local pollution case not filtered by the dynamic or chemical filters, and the post-treatment made by the PI.

[Figure]

All the data used in the paper results ruled out any risk of local contamination.

Another type of manual intervention by the PI concerns instrumental alert messages. This mainly concerned CPCs which require a filled water tank. When it is dry, error messages appear and the data must not be used. We are working on automatic detection methods for non-physical data and instrument alerts but they are not yet reliable and data supervision by the PI is still necessary.

> *Similarly, there is an automatic flags calculation system (in section 4.2.3) without any details about how it is done.*

You're right, it's not clear whether the automatic flag indicated in section 4.2.3 corresponds to the dynamical filter described in 3.3.

The sentence has been modified in the paper section 4.2.3 as follow:

"Therefore, the dynamic flag calculation described in section 3.3 has been set up automatically  on the project's web servers, indicating for each measurement whether it is likely to have been polluted by stack emissions."

Minor comments:

> L33-38: This part is a bit disconnected from the previous and later texts. I'd recommend first adding 1-2 sentences to explain why the Southern Ocean is important, to connect with the previous texts (that observations over oceans are limited). Then after the 3 pathways of BDC, maybe add some texts to

explain why these pathways are related/important to Southern Ocean. Now the transition from this part to L38 "Although two stations" is confusing to me.

L41-43: this also needs some transitions between the two sentences. The first sentence "The observation of …" talks about barriers, then suddenly the next sentence talks about tropospheric transport.

You're right on both counts. We've changed this part and moved it after the sentence on the biomass fires. We think it makes more sense.

The paragraph is now written as follows:

"The penetration of smoke-related compounds in the stratosphere is thought to be more frequent in a warming world, and depends on pyroconvection mechanisms (Fromm et al., 2000) as well as on the tropopause which acts as a dynamical barrier. Therefore, the climatic impacts of such plumes have to be assessed in this poorly documented part of the world. Monitoring of atmospheric changes in the free troposphere and stratosphere over the Indian Ocean is sorely lacking, with only two NDACC stations (https://ndacc.larc.nasa.gov) located on Reunion Island (21 S) and Kerguelen Island (49.3 S). The spatial distribution of key radiatively active trace gases, such as ozone in the stratosphere, is largely affected by the Brewer-Dobson circulation (BDC) involving three latitudinal regions from the equator to the pole (Butchart, 2014): (i) the tropical stratosphere reservoir, (ii) the strong mixing mid-latitude surf zone and, (iii) the polar vortex. Those regions are separated by a permanent subtropical dynamical barrier and a winter polar barrier. Chemistry-climate and climate models predict a strengthening of the BDC, especially within its shallow branch, due to the increase of greenhouse gases (Abalos et al., 2021). Observing ozone over a wide area of the Indian Ocean with repeatable trajectories will enable robust characterization in the different regions separated by these dynamic barriers."

L197: do the 4 cylinders have different concentrations? What are the concentration ranges of calibration standards for these gases as well as for other gases and aerosols?

These calibration cylinders only concern the greenhouse gases (CO, CH4, CO2). It corresponds to the WMO and IR ICOS international standards. As explained before, we added in the text some information about this procedure of calibration and cylinder gas concentration:

"For calibration, we use 4 gas cylinders calibrated at LSCE according to WMO international scales. The calibration scale has a concentration range from 396 to 472 ppm for CO2, from 1760 to 1960 ppb for CH4, and from 25 to 374 ppb for CO. A calibration sequence is performed every month, or after an interruption of the analyzer."

Note that the calibration procedure for O3 and NOx uses a multigas calibrator as mentioned at line 209.

L222: should add numbers or a plot (in the supplement) to show the uncertainty.

The monthly average of PM2.5 mass concentrations of the 3 OPC-N3 were compared over one year (June 2021 to June 2022).

Apart from OPC-N3 N°3 in October 2021 which underestimates the PM2.5 concentration by 20 μg/m3 compared to the other two instruments, the difference between the three instruments is less than 10 μg/m3. The average relative errors are 1.7% between OPC-

N3 N°1 and OPC-N3 N°2, 9% between OPC-N3 N°1 and OPC-N3 N°3 and 7.3% between OPC-N3 N° 2 and OPC-N3 N°3.

In the paper we have used the data of the OPN3 N°1 which is close to the data of the OPC-N3 N°2 (see below).

These explanations and the figure have been added as supplement.

[Figure]

Section 3.4.1: what is the precision and detection limit? How are the cloudy days processed?

As mentioned in the text, the installation on Marion Dufresne of the CIMEL 318-T photometer follows all the standards of processing and calibration as other photometers in the AERONET network. For cloud screening, we follow the same process as the one described in Giles et al. (2019) for AERONET Version 3 photometers. The uncertainties of the AOD cloud screening data are equivalent to those of other photometers in the network, estimated to be 0.02 for ultraviolet channels (340 nm) and 0.01 for the channels (see the aforementioned paper by Giles and others such as Holben et al. 1998, Eck et al. 1999, Holben et al. 2006). The quality of the radiances obtained is still under analysis, and the first retrievals combining AOD and sky radiance (standard AERONET inversion) from the boat will be the subject of a new publication coming soon. All the references are in the paper of Gilles.

L436-437: in the publicly available dataset, are data filtered by both NO and CO methods?

Yes, all CO and NO filters are available in the public data. These two filters (as well as the dynamic filter) were used for the results of the article.

Figure 11: why so many CN data are removed for the regions around [35S,75E] to [40S,80E] compared to other gases (e.g. Figure 9 same regions)?

This is true. The difference between trace gas and aerosols number plotted could have three explanations:

1- Aerosol measurements were stopped in 2022, so we don't have all the campaigns that were used for gases.

2 - Processing by the PIs may exclude data on the basis of an instrumental malfunction: e.g. if the water tank of the SMPS and the CPC was dry, the instruments gave non-physical data (see previous answers).

3 - As previously mentioned, the post-processing of aerosol data contained python coding errors. This is the case for the area you mentioned.

Section 7: I randomly checked the CO, CO2 data, but there is no latitude, or longitude info in the dataset, why?

We have made a verification by uploading the CO data on the aeris-data portal, and the latitude and longitude are present on the file. For example, the L1 data for CO are given as follow on the csv file:

**Site;SamplingHeight;Year;Month;Day;Hour;Minute;DecimalDate;co;Stdev;NbPoints;Flag;InstrumentId;QualityId;InternalFlag;AutoDescriptiveFlag;ManualDescriptiveFlag;**Latitde;Longitude**;Altitude-AMSL;Altitude-AGL**

It is true that the raw data (L0) does not include the position: these files only give the raw outputs of the instruments. To plot this raw data according to the ship's position, you must also retrieve the position file "INS-POSITION MAP-IO LEVEL 1" from the aeris-data portal.

In the new version of the paper, we have detailed the data files used and their location in section 3.1 and table 1.

Specific comments:

Thank you very much for pointing out these spelling errors. We really appreciate your help.

Keywords: missing

Keywords are not requested by the ESSD journal.

L2: consider using 'with' instead of 'thanks to'

It has been corrected.

L3: observation – observational

It has been corrected.

L4: ships – the ship [if there was only one ship]

It has been corrected.

L10: the journal usually requires a sentence to include the data source/link, please check with the journal.

This is true. We added the sentence:

"The meteorological MAP-IO dataset is publicly available at https://www.aeris-data.fr/catalogue-map-io/ (atmospheric data) and at https://www.seanoe.org/data/00783/8950 (phytoplankton data)."

Abstract: I recommend adding 1-2 sentences at the end to mention the potential implications of this dataset or science questions that can be explored with the data.

This is right. We added the sentences:

"The multi-year rotations over the Indian Ocean will enable us to assess the trends and seasonal variability of phytoplankton, greenhouse gases, ozone, and marine aerosols in a sensitive and poorly documented climatic region. Without being exhaustive, MAP-IO should make it possible to better understand and assess the biological carbon pump, to study the variability of gases and aerosols in a region remote from the main anthropogenic sources, and to monitor the transport of stratospheric ozone by the Brower-Dobson circulation."

L12: remove 'probably'

It is done.

L17: what is 'earth climate budget'?

Budget has been deleted.

L18: estimates – estimated

Thanks, it has been corrected.

L25: as well – such as

Thanks, it has been corrected.

L27: add what WMO stands for

We have changed the WMO acronym by the World Meteorological Organization.

L61: have to – should

Thanks, it has been modified

L90: A – The

Thanks, it has been corrected.

L91: a third one – the third one

Thanks, it has been corrected.

L100: The rest of – During the rest of

Thanks, it has been corrected.

L104: is to carry – carries; integrated – integrates

Thanks, both have been corrected.

L105: remove 'a focus of'; interest for -  interest in

Thanks, both have been corrected.

L106: to establish – of establishing

Thanks, it has been corrected.

L120: at – in; move 'by July 2023' to the end of the sentence

Thanks, both have been modified.

L129: pollution -  contamination?

Yes, the word contamination is better. Thanks.

L130: remove 'located'

Thanks, it has been corrected.

L211: add 'which is' before 'able to'

Thanks, it has been modified.

L330: it – is

Thanks, it has been corrected.

L448: 90% of the 2021-2023 measurement days or of all the days during 2021-2023 (365 days/year)?

Yes it was uncleared. It was on the measurement days at sea. We modified the sentence as:

"The reactive gases instruments worked on average 90% of the measurement days at sea between 2021 and 2023."

L470: what evidence supports the statement 'This corresponds to…'?

There are no CO emissions from the ocean. As these are not occasional concentration peaks which could have been attributed to local pollution due to the ship's activities, we assume that the origin is large-scale transport.  The sentence was rewritten as follows:

"As there are no CO emissions over the ocean, these high concentrations are attributed to long-range transport; usually of biomass burning plumes."

L475: what range of quantiles, 25-75%?

Yes, we have written quartiles that mean quantiles 25-75%. We have modified the paper to be precise.

L476: correctly – normally

Thanks, it has been corrected.

Figure 9: should explain in the figure legend what the texts above the dates (such as 'OP1-2021') represent.

We think this comment is intended for Figure 8. We added these explanations in the legend of figure 8 and in the legend of the modified figure 10, as : "The names of the campaigns and their respective dates have been entered on the x-axis."

Figure 10: (1) x-axis represents month? (2) what are the ranges of quantiles?

It was months but this figure has been modified to give campaigns on the x_axis.

Figure 13: missing x-axis label

**Referee 2:**

The paper presents a multidisciplinary MAP-IO dataset that includes ship-based measurements of atmospheric gases, aerosols, clouds, and phytoplankton over the pristine Southern Ocean. The dataset is of high quality and holds immense potential for research and model evaluations/parametrizations. It is highly necessary, given the sparse availability of observations for such regions. The article provides ample information regarding the instruments used in the measurements and the collection of the datasets, which has already made it quite lengthy for a data descriptor paper. The initial results are interesting and highlight the manifold potential of the dataset collection. However, the article lacks some necessary information required for a data descriptor paper to ensure that users can fully understand and use these datasets (when they are public). I recommend publication of the article after addressing the following comments.

We would like to thank the reviewer for its work, which we believe has significantly improved the article. We have done our best to answer its questions and recommendations.
We would like to draw your attention to an error in the post-processing of the SMPS and CPC data which has impacted the results presented in figures 10, 11, 12 and table 2.
The new results do not contradict the previous version of the article, but are sufficiently different to partially modify the section 5.3.2. The instrument's post-processing algorithm has been checked several times and we are now confident in the reliability of the results. We sincerely apologize for this computation error.

1. As this is a data descriptor paper, I suggest the authors add a table that includes the format of file names for all datasets described in the article, along with a short description of the parameters stored in those files. This approach will make it much easier to locate the files in the archive, once it is made open to the public. This information can be incorporated into additional columns in Table 1.

Thank you for this remark.
We agree that the presentation of data and their localization must be improved. The choice adopted was to clarify the name and variables directly on the data centers. We prefer this solution because both AERIS and SEANOE can evolve and it is preferable that the description of the format and the variable names are on the download site rather than fixed in an article because it is potentially obsolete in a few years.
According to your recommendation, we have also included in the article, the access site for each data used in the article and the corresponding block to download them (table 1). For phytoplankton data from the SEANOE data center, the explanation of the available variables has been added in the Device section via a link to the description of the BODC F02 groups.

For data from the AERIS data center, information on the name and units of the variables is now described in the summary part or in the header of the csv or text file. We hope this is clearer now.

2. While reading the manuscript, the first thing I searched for was a link to the datasets. I found multiple links within the article with their own set of functions, although not necessarily intended to provide the data. For example, https://www.osureunion.fr in Section 4.2. Additionally, the link to the SEANOE datacenter (https://doi.org/10.17882/89505) is provided in the data availability section, but its description is missing in the manuscript.

Since this paper presents multiple datasets that are relevant to researchers from various interdisciplinary fields, I suggest the authors consolidate all the links in one section (e.g., Section 4.2 of the manuscript) and clearly state where readers can download the datasets once it is published and opensource.

Again, we agree, it's too confusing in the article. Firstly, there is confusion between the local base of direct access to data offered to PIs (with restricted access), with the base open to the data centers.
The OSU-R site is not the site for downloading data by users but the laboratory takes care of maintaining the operation of the instruments and collecting the data and transferring it to the data centers. It offers direct download access to IPs but this should not be stated in this article.
There are two unique sites for downloading the data: 1) SEANOE (SEA sciNtific Open data Edition) for the marine observations (phytoplankton): http://en.data.ifremer.fr/Submit-Archive-data/SEANOE, and 2) AERIS for the atmospheric measurements: https://www.aeris-data.fr/en/welcome-2/
These two centers then take care of transferring the data to international centers such as AERONET, NDACC, ECAC etc..
This development is in progress but not yet available. The section 4.2 has been modified to clarify all of this.

The following paragraph (section 4.2) has been deleted.
*"All data and services (FTP, HTTP, SQL) offered to the end user are within a secure area (DMZ) of the University of Reunion and maintained by the OSU-R (https://www.osureunion.fr, last access: 8 November 2023). Data access for users is provided through secured FTP or secured MySQL as a pulling mode. Also, OSU-R pushes data to archiving centers such as AERIS*
*(https://www.aeris-data.fr, last access: 8 November 2023) after formatting. OSU-R is in charge of long term data archiving. "*

The section 4.2.4 has been modified as follow:
*"The MAP-IO web servers provide PIs with secure access to data via the FTP protocol. All data retrieved from the Marion Dufresne instruments and offered to the PI via the project's web servers are also archived in real time in a MySQL database at OSU-*

*Reunion. Raw atmospheric data (level 0) are transferred daily to the AERIS data center. The PIs are responsible for data analysis and validation according to quality protocols defined by international standards.*

*Within a year, all acquired data will have been validated and post-processed by the PIs (to level 1.5 or 2) and transferred to international data centers such as AERIS for the atmosphere ([https://www.aeris-data.fr/catalogue-map-io](https://www.aeris-data.fr/catalogue-map-io) , last access: November 8, 2023) and SEANOE ({https://www.seanoe.org/data/00783/89505) for the ocean.*

*In the near future, these data centers will then be responsible for transferring the MAP-IO data to international centers such as EBAS ([https://ebas.nilu.no/](https://ebas.nilu.no/) for in-situ aerosols), AERONET ([https://aeronet.gsfc.nasa.gov](https://aeronet.gsfc.nasa.gov) for photometer) or NDACC ([https://www-air.larc.nasa.gov/missions/ndacc/](https://www-air.larc.nasa.gov/missions/ndacc/) for mini-SAOZ)."*

Also the Figure 2 has been changed to be cleared by pointing in particular to the data centers.

3. I noticed that the data files are in CSV format and the parameters lack basic attributes, such as units and long names. While some variables are standard and straightforward to understand just by their names, others, like "filtre_vaisala_d0.55945_w0.5590_ff4," are very complex and would therefore be confusing for users unfamiliar with in-situ terminologies. I recommend using the NETCDF format to store the variables and following the CF conventions for attributes, which makes it easier to incorporate them into climate models and existing software and tools that support CF conventions. An alternative fix could be to include all attributes in a table within the manuscript. However, I strongly recommend using the NETCDF format. Further, the authors should add another section to the manuscript describing the data specifications (information on all variables), which is currently missing.

As explained above, we rely on French data centers. We have completed the metadata to make explicit the name and unit of each variable used in the article.
We agree with you that it would be useful to have the option of using the NETCDF format.
A request has been made to the AERIS data center, and this option is currently being examined. We expect to be able to offer this type of download within a few months. Initial tests have been carried out, but the procedure is still too complex to be broadcast.

Minor comments
1. Include a data repository link in the abstract of the manuscript once the data is openly available.
We added the SEANOE and AERIS data links in the new abstract.

2. The team has done a commendable job in designing the website http://www.mapio.re/, particularly the near real-time graphical representation of the datasets. However, there is no English translation available in the wiki section and some other links within the website. For a future update, I highly recommend the authors to

include an English option as well, making it easily accessible to a broader audience. Nevertheless, with this publication, this issue should be addressed.

The reviewer is right. The English version of the website was in our plans.  It is now done.

3. Section 3.3.4. What is the number of bins considered in the size distributions?

**We use 80 bins. We added this information in the text.**

4. In lines 476-477, you mention that the average aerosol number concentration during MAYOBS in Sept. 2021 is about 1200 cm−3. However, in Fig. 10 of the manuscript, there are no observations corresponding to September 2021. Perhaps the x-axis scale of FIg. 10 needs to be adjusted.
Yes, it was a problem of x-axis. We have changed the abscissa axis of Figure 9 depending on the campaigns as was done for Figure 8.

5. In line 508, change "sursaturation" to "supersaturation".
Thanks, it has been changed.

6. In Figure 12 caption, is it "root mean square deviation" or "interquartile range"?

The vertical lines in Figure 12 represent the quartiles (quantiles 25-75%) computed for each mode and for the bin corresponding to the mean diameter of Table 2. The RMSE per mode is calculated for each lognormal distribution in table 2.